# Natural variation in *Glume Coverage 1* causes naked grains in sorghum

Peng Xie [1,2✉], Sanyuan Tang[1], Chengxuan Chen[1,2], Huili Zhang[1], Feifei Yu[1], Chao Li[1], Huimin Wei[1,2], Yi Sui[3], Chuanyin Wu[3], Xianmin Diao [3], Yaorong Wu [1✉] & Qi Xie [1,2✉]

One of the most critical steps in cereal threshing is the ease with which seeds are detached from sticky glumes. Naked grains with low glume coverage have dramatically increased threshing efficiency and seed quality. Here, we demonstrate that *GC1* (*Glume Coverage 1*), encoding an atypical G protein γ subunit, negatively regulates sorghum glume coverage. Naturally truncated variations of GC1 C-terminus accumulate at higher protein levels and affect the stability of a patatin-related phospholipase SbpPLAII-1. A strong positive selection signature around the *GC1* genic region is found in the naked sorghum cultivars. Our findings reveal a crucial event during sorghum domestication through a subtle regulation of glume development by GC1 C-terminus variation, and establish a strategy for future breeding of naked grains.

[1] State Key Laboratory of Plant Genomics, Institute of Genetics and Developmental Biology, The Innovative Academy of Seed Design, Chinese Academy of Sciences, 100101 Beijing, P. R. China. [2] University of Chinese Academy of Sciences, 100049 Beijing, P. R. China. [3] Institute of Crop Sciences, Chinese Academy of Agricultural Sciences, 100081 Beijing, P. R. China. ✉email: pxie@genetics.ac.cn; yrwu@genetics.ac.cn; qxie@genetics.ac.cn

The hulled grains of wild cereal species can prevent the grains from destruction by fungi and insects to guarantee propagation and survive in harsh environments[1]. However, hulled grains in which seeds are tightly wrapped by a pair of tenacious glumes are a huge obstacle to threshing process in human agricultural production. Modern high-intensity mechanical threshing can easily break the embryo of hulled seeds, affecting grain quality and germination rates and decreasing the market value. The inefficient manual threshing is not a viable large-scale alternative as it is labor-intensive and difficult even for practiced farmers[2]. Therefore, the emergence of naked grains, which have seeds that can be conveniently separated from glumes, has been a highly important step in cereal threshing.

Sorghum is one of the first major crops cultivated by ancient humans[3], and naked grains are common among available accessions, due mainly to variation in glume size (hence referred as "glume coverage"). In the modern agricultural practice, we noticed that the hulled sorghum led to huge difficulties for automated planting and resulted in a great production loss in the field and inefficient postharvest threshing. In contrast, the naked sorghum varieties are more attractive to farmers due to a much higher efficiency in mechanized planting, threshing and processing. As early as 80 years ago, scholars used glume coverage as a key reference character in the classification of sorghum subspecies[4]. However, the genetic basis of glume coverage in sorghum remains unknown.

G protein complex is a trimer of proteins with GTP binding and hydrolysis activity, which is composed of α, β, and γ subunits. G protein plays a 'molecular switch' role in signal transduction from extracellular to intracellular environments, affecting multiple physiological processes in plants[5,6]. Both in plants and mammals, phospholipases constitute a key regulatory part of the G-protein cycle and are involved in injury, pathogen defense and cell growth[7,8]. The phospholipase A-II (PLAII) functions as a hydrolase that can catalyze the second acyl bond on phosphoglycerol lipids to produce lysophospholipids and free fatty acids. Both two products play an important role in various cellular signaling processes by generating second messenger molecules[9]. However, the relationship between Gγ subunit and phospholipases in regulating plant cell development remains unknown.

In this study, we identify a major gene, GC1 (Glume Coverage 1), which encodes an atypical Gγ-like subunit in controlling glume coverage in sorghum. And we reveal that GC1 regulates the degradation of patatin-related phospholipase II-1 (pPLAII-1). The naturally truncated alleles can be beneficial for the naked grain breeding in sorghum.

## Results

### GWAS and map-based cloning of GC1 locus.
We firstly analyzed the morphology of glume coverage in 915 diverse sorghum accessions, and found that all wild sorghum accessions show seeds absolutely enveloped by a pair of tenacious glumes (grain hulled) while nearly 60% of sorghum cultivars show seeds with low or very low glume coverage (grain naked) during domestication (Supplementary Fig. 1a and Supplementary Data 1). Also, we noticed a significant positive correlation ($R^2 = 0.98$) between glume coverage and glume length (Supplementary Fig. 1b). With the decrease of glume coverage, the threshing efficiency was obviously increased (Supplementary Fig. 1c). Then we conducted a genome wide association study (GWAS) using a sorghum association panel (SAP, consisting of 352 inbred lines)[10] and phenotype data collected from mature sorghum grains with five defined classes of glume coverage (Fig. 1a). Using these data, we identified three major loci ($P$-value $< 10^{-6}$) which were located on chromosomes 1, 2, and 3 (Fig. 1b and Supplementary Data 2). In addition, we also mapped a major locus, GC1, located on the long arm of chromosome 1 in an $F_6$ population from a cross between the hulled, hard-threshing line SN010 and the naked, easy-threshing line M-81E (Supplementary Fig. 2a). Comparative genomic analysis showed that GC1 was collinear to the locus detected on chromosome 1 identified through GWAS (Supplementary Fig. 2b and Supplementary Data 2). These results suggest that GC1 is a main-effect and stable locus linked with glume coverage in sorghum.

Next, to fine map the GC1 locus, we continued to develop two offspring generations which included 1678 plants in total by flanking marker-assisted selection (MAS) (see "Methods" section). Using 40 screened recombinants, the GC1 locus was ultimately narrowed down to a ~58 Kb genomic interval, in which there are five annotated genes. Only the last three of these five genes were found to be expressed in panicle tissues, suggesting a possible function in spikelet development. Then we analyzed all genetic variations including 257 SNPs and 103 indel polymorphisms across the GC1 region in 57 sorghum inbred lines (Supplementary Data 3). An indel site Indel-62915822 located inside the fifth exon of the third gene showed a leading association peak with glume coverage ($P = 1.4E-33$) (Fig. 1c). Thus, this gene (Sobic.001g341700) was considered to be the candidate gene of GC1.

To confirm whether sequence polymorphisms of GC1 are responsible for glume coverage in sorghum, we sequenced the GC1 coding region in 482 sorghum accessions. All genotyped sorghum accessions were classified into five symbolic GC1 haplotypes, which include one wild type and four truncated types. Four vital malfunctional mutations were identified in the fifth exon of GC1: a GTGGC insertion (referred as gc1-a allele), a G deletion (gc1-b), a C-A SNP mutation which obtained a stop codon (gc1-c) and a 165 bp insertion (gc1-d) (Fig. 1d and Supplementary Fig. 3). Haplotype-based association analysis showed the natural variations at +4158 of GC1 were the most significantly associated with glume coverage ($P = 5.27E-14$). Linkage disequilibrium (LD) analysis further indicated that the leading three association signals (at +4151, +4158, and +4285) within the fifth exon region were attributable to the strong LD ($r^2 > 0.9$) (Fig. 1e). However, none of these variations showed a powerful association signal with grain yield-related traits.

### Gγ-like subunit encoded by GC1 is crucial for glume coverage in sorghum.
Sorghum G protein contains one Gα, one Gβ, and two typical and three atypical Gγ subunits, which is similar to G proteins in rice[11] and millet (Supplementary Fig. 4a). GC1 encodes a peptide product of 198 amino acids with an atypical G protein γ subunit-like (Gγ-like) domain located in the N-terminus (referred as GC1-G) and a predicted transmembrane domain (GC1-T) (Supplementary Fig. 3). We searched for homologs of GC1 in other cereal species and found GC1 was an ortholog of the rice GS3[12], with 50.86% protein sequence similarity. The N-terminal Gγ-like subunit is highly conserved (peptide similarity > 80%) while the C-terminus shows a high level of variation (peptide similarity < 10%) among the homologs in seven crops (Supplementary Fig. 4b). To verify the function of variant GC1 alleles, we constructed a pair of near isogenic lines (NILs) for the GC1 locus (Fig. 2a). Compared with NIL-GC1, NIL-gc1-a produced a premature stop codon at amino acid position 137 of the C-terminal (Supplementary Fig. 3), and exhibited much lower glume coverage with a huge reduction in glume length and slightly shorter glume width, resulting in a more than 60% threshing rate increase (Fig. 2b, c and Supplementary Fig. 5). This further indicates that the naturally truncated variations in GC1 may confer lower glume coverage in sorghum.

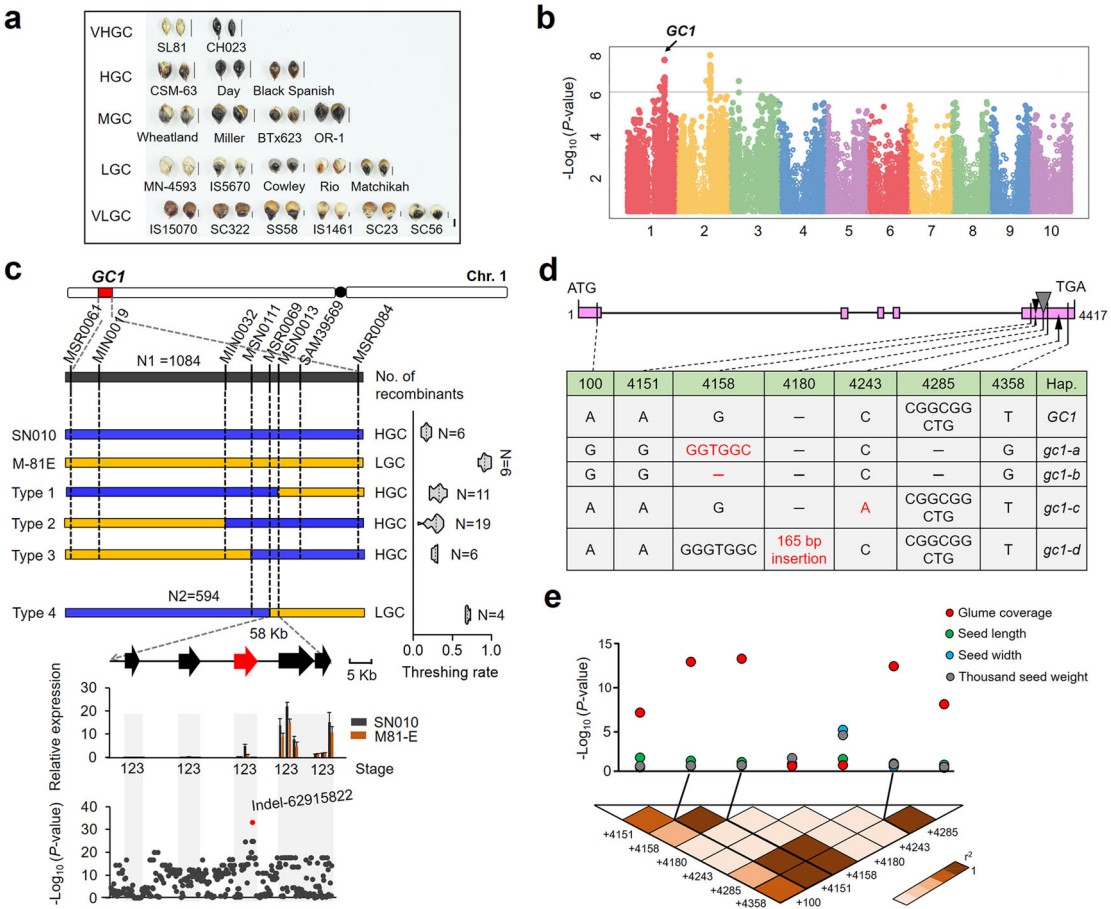

**Fig. 1 Natural variations in *GC1* are highly associated with glume coverage in sorghum. a** Five defined glume coverage degrees in diverse sorghum accessions. The corresponding thin line near each spikelet means quantitative glume length. Bar = 2 mm. VHGC very high glume coverage, HGC high glume coverage, MGC moderate glume coverage, LGC low glume coverage, VLGC very low glume coverage. **b** Manhattan plot from GWAS analysis of glume coverage in the natural SAP sorghum population. The gray horizontal line indicates the Bonferroni-adjusted significance threshold ($P = 1E-06$). **c** Map-based cloning of *GC1* locus through the 1678 offspring plants of RHLs (see "Methods" section). The violin plot represents threshing rate. The dotted line in the middle is the median. The thin rectangles mean genomic region. The thick arrows depict five annotated genes. The red color represents the candidate gene. Relative expression of the five annotated genes at S1 (primordia), S2 (young panicle), and S3 (mature panicle) stages in two parental lines. Data are mean ± s.e.m. $n = 3$ biological replicates. Association analysis between the genetic variations and glume coverage in 58 sorghum inbred lines. The red dot shows the leading association signal on the fifth exon of the third candidate gene. **d** Nucleotide polymorphisms within the *GC1* coding region of 482 sorghum accessions. Four detected malfunctional variations (highlight in red) in the fifth exon of *GC1*: a "GTGGC" insertion (*gc1-a*), a "G" deletion (*gc1-b*), a "C-A" substitution (*gc1-c*) and a 165 bp fragment insertion (*gc1-d*). Hap. haplotype. **e** *GC1*-based association mapping between the seven variations and four spikelet related traits in 188 sorghum accessions. LD analysis between the seven causal sites in the coding region indicates the linkage association signals. *P*-values were determined by two-tailed unpaired *t*-test. The three leading variant sites (+4151, +4158, and +4285) show highly association signals with strong LD which are highlighted by black lines. Source data are provided as a Source Data file.

To deeply explore the molecular function of *GC1*, we generated transgenic plants expressing the Myc-tagged GC1 driven by a *ubiquitin* promoter in Wheatland, a sorghum recipient inbred line which harbors the wild *GC1* allele and has moderate glume coverage. The *GC1* RNA levels were tested in five independent *GC1*-overexpressing (*GC1-OE*) plants of the $T_0$ generation (Supplementary Fig. 6a). In contrast to the Wheatland control, *GC1-OE* plants exhibited slightly shorter glumes with lower glume coverage, and a 13% increase in threshing rate. Additionally, we generated a *GC1-Knockout* (*GC1-KO*) mutant (Supplementary Fig. 6b) in Wheatland by genome editing technology[13]. Three independent $T_2$ families for *GC1-KO* homologous mutants with separated carriers were investigated. Surprisingly, compared to the wild type, knocking out *GC1* resulted in much longer and hard-threshing glumes with an all-enveloped glume coverage trait (Fig. 2d–f). These results suggest that *GC1* may negatively control glume coverage in sorghum.

The *GC1-KO* shows high glume coverage, which is different to what we found in *gc1-a*. It indicates that the *gc1* alleles may still express truncated proteins to affect glume development. Then, we generated a Myc tagged gc1 (gc1-OE with amino acid position from 141 to 198 of GC1 deleted but containing complete Gγ-like domain and transmembrane region) construct driven by a *ubiquitin* promoter and transformed into Wheatland (Supplementary Fig. 6a). We found that all of the *gc1-OE* transgenic plants of the $T_0$ generation showed much reduced glume length with lower glume coverage than both Wheatland and *GC1-OE* plants, with a 55% increase in threshing rate in *gc1-OE* compared to Wheatland (Fig. 2d–f). Together, these results indicate that GC1, particularly the C-terminus truncated version, acts as an important negative regulator of glume coverage in sorghum.

To confirm our observation of Gγ-like subunit function in glume development, we performed genetic analysis of SiGC1, a GC1 homolog in millet (*Setaria italica* L.) and with a 78.3%

 3

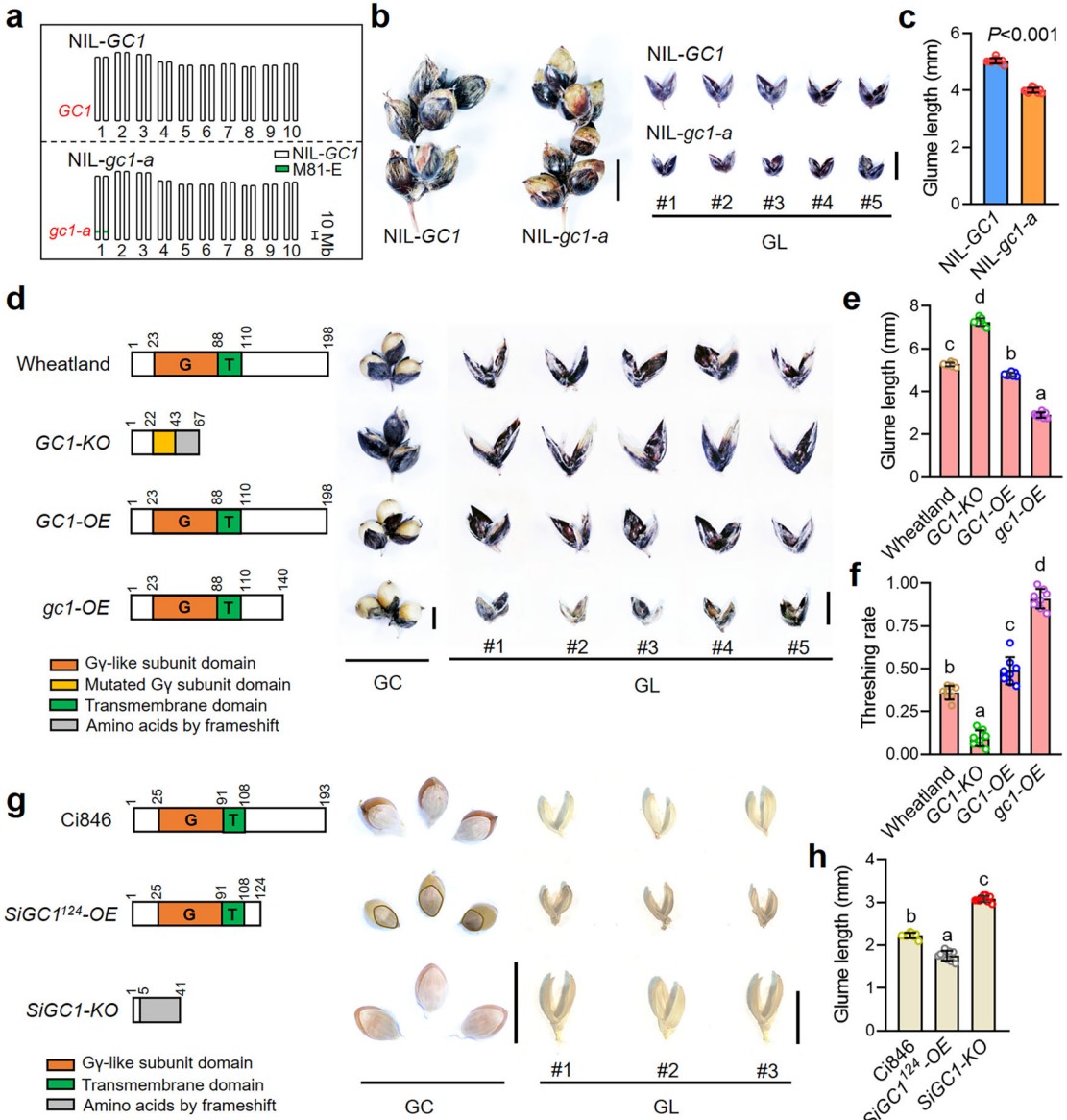

**Fig. 2 The Gγ-like subunit negatively controls glume coverage in sorghum and millet. a** Genetic background of NIL-*GC1* and NIL-*gc1-a* lines based on the self-crosses process derived from the cross between SN010 and M-81E. The green box in the long arm of chromosome 1 indicates the homozygous 58 Kb *GC1* region derived from M-81E (*gc1-a/gc1-a*) (see "Methods" section). **b** Morphology of mature spikelet and glumes in NIL-*GC1* and NIL-*gc1-a*. Bar = 5 mm. **c** Statistics of glume length in NIL-*GC1* and NIL-*gc1-a*. Phenotypic data are mean ± s.e.m. *n* = 10 biological replicates. *P*-values were determined by two-tailed unpaired *t*-test. **d** Schematic diagram of peptide structure, mature spikelet and glumes in Wheatland (wild type of *GC1*), *GC1-KO* mutant, *GC1-OE*, and *gc1-OE* lines. Bar = 0.5 cm. **e**, **f** Statistics of glume length and threshing rate in the plants shown in **d**. Data are mean ± s.e.m. *n* = 8 biological replicates. **g** Schematic diagram of peptide structure, mature spikelet, and glumes in Ci846 (wild type of *SiGC1*), *SiGC1^124-OE* line and *SiGC1-KO* mutants. Bar = 0.5 cm. The glume outline of *SiGC1^124-OE* is highlighted by yellow lines. **h** Statistics of glume length in the plants shown in **g**. Data are mean ± s.e.m. *n* = 9 biological replicates. G, G protein γ subunit domain. T transmembrane domain, GC glume coverage, GL glume length. *P* values in **e**, **f**, **h** were determined by one-way ANOVA with Tukey's multiple comparisons test. Source data are provided as a Source Data file.

protein identity with GC1 (Supplementary Fig. 4b). Millet has closely related spikelet morphology (Supplementary Fig. 7a) and a high genome similarity to sorghum[14]. We synthesized *SiGC1^124* mimicking the sorghum truncated allele, which encodes a peptide of 124 amino acids and contains the entire Gγ-like subunit and transmembrane region. A Myc tagged *SiGC1^124* with *ubiquitin* promoter construct was created and transformed into Ci846, a recipient millet inbred line with full-length *SiGC1* allele (Supplementary Fig. 7b). Three independent *SiGC1^124-OE* transgenic plants in the T₀ generation were identified and showed a dramatic reduction in glume length with low glume coverage (Fig. 2g, h). Furthermore, we generated a *SiGC1-KO*

mutant in Ci846 by the similar genome editing system (Supplementary Fig. 7c). As expected from the results in sorghum, *SiGC1-KO* mutant had enlarged glume size with an ~30% glume length increase compared to the control Ci846 (Fig. 2g, h). Based on these genetic evidences in both sorghum and millet, we conclude that the truncated Gγ-like subunit acts as a key negative regulator of glume coverage.

**The truncated C-terminus of GC1 allows increased protein accumulation.** To verify that the C-terminus truncated proteins are expressed in the transgenic sorghum plants in which we

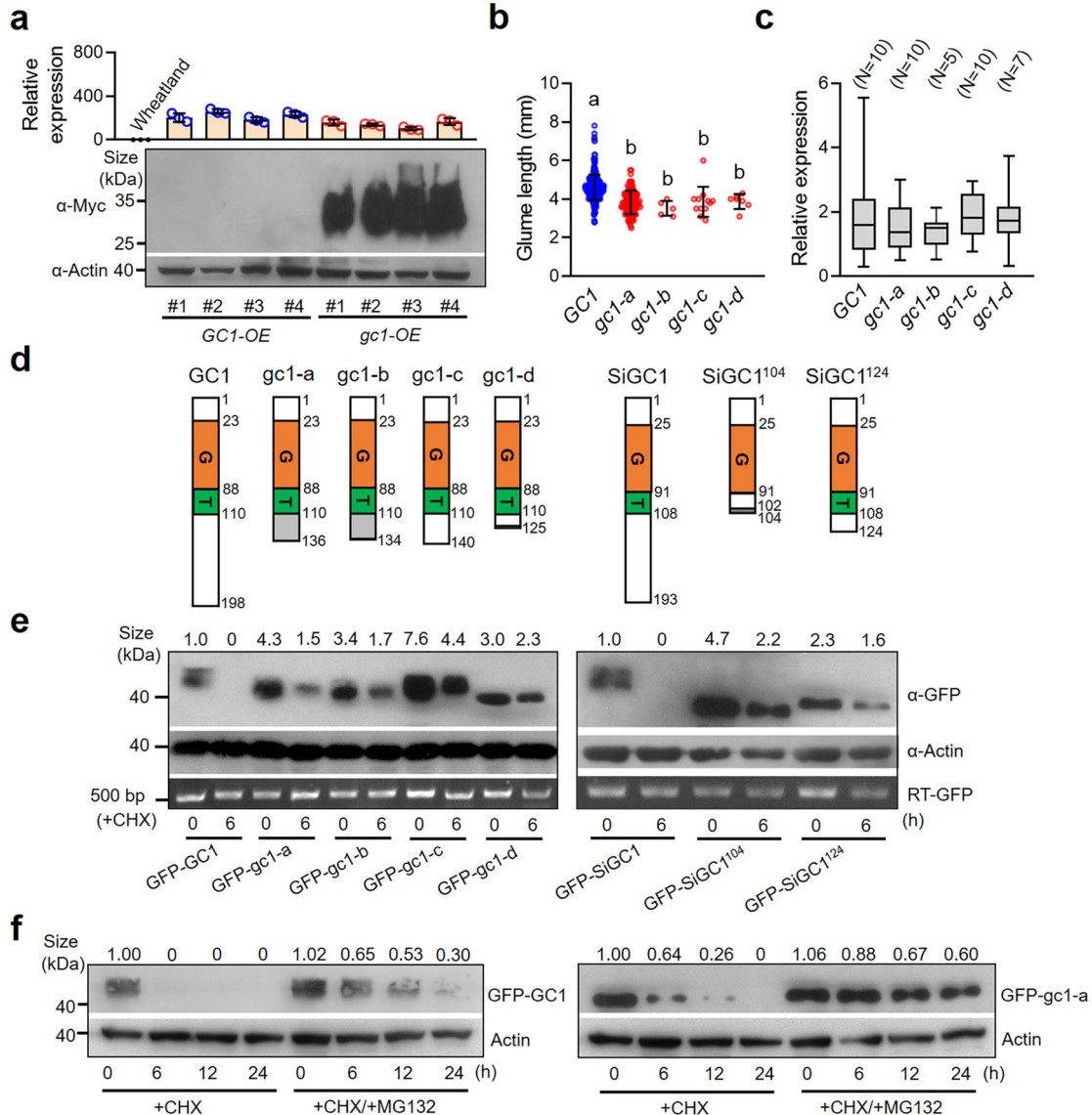

**Fig. 3 The C-terminal truncated gc1 confers low glume coverage by accumulating higher protein level. a** Immunoblot analysis of Myc tag fused GC1 and gc1 proteins extracted from the young panicles of transgenic *GC1-OE* and *gc1-OE* sorghum plants, respectively. The relative expression of *GC1* and *gc1* in transgenic plants compared to Wheatland were detected by qPCR. **b** Glume length evaluation of five *GC1* haplotypes among 482 sorghum accessions. Data are mean ± s.e.m. **c** Comparison of *GC1* expression level among the five *GC1* haplotypes. The two ends of the box plot and the upper, middle, and lower box lines represent the upper edge, lower edge, median and two quartiles of values in each group. Statistical significance was determined by one-way ANOVA with Tukey's multiple comparisons test in **b** and **c**. **d** Schematic diagram of peptide structure of natural or synthetic GC1, gc1-a, gc1-b, gc1-c, gc1-d, SiGC1, SiGC1[104], and SiGC1[124] proteins. G, G protein γ subunit domain. T, Predicted transmembrane domain. The gray color shows the out-frame amino acids. **e** GFP tag fused with various deletion-based GC1 and SiGC1 proteins showed in **d** were transiently expressed in *N. benthamiana* leaves. Protein levels were subsequently analyzed by western blotting with anti-GFP antibody after 30 μM CHX treatment for 0 and 6 h. The expression of GFP tag from tobacco leaves were detected by RT-PCR. **f** The protein accumulation of GFP-GC1 and GFP-gc1-a extracted from tobacco leaves after treatments with CHX (30 μM) or CHX (30 μM) + MG132 (50 μM) for 0, 6, 12, and 24 h were analyzed by western blotting with anti-GFP antibody. Plant actin was used as the loading control. CHX cycloheximide. The numbers on the blot bands indicate the relative protein quantification by determination of grayscale value. Three independent experiments were performed in **e** and **f**. Source data are provided as a Source Data file.

observed reduced glume coverage, we first analyzed the protein abundance in *GC1-OE* and *gc1-OE* plants with similar RNA levels. Western blot results showed that a large amount of Myc-gc1 accumulated while Myc-GC1 could barely be detected in the young panicles of transgenic sorghum plants (Fig. 3a). Then we returned back to analyze the protein status of different naturally allelic effects on glume coverage and found the truncated gc1-a/b/c/d with short C-terminus (short tails, hence after) were associated with significantly shorter glumes than that with the full-length GC1 (long tail) (Fig. 3b). This may imply that the long tail of GC1

represses the inhibitory function of N-terminal Gγ-like domain while short tails relieve the inhibitory effect of the N-terminus. We also observed no significant differences in gene expression among the five *GC1* alleles (Fig. 3c), suggesting that divergent glume coverage affected by GC1 tails is possibly due to variation in protein level rather than in transcriptional level.

To further verify this hypothesis, we transiently expressed vectors containing a GFP tag fused with each of eight deletion-based GC1 and SiGC1 proteins in *Nicotiana benthamiana* leaves (Fig. 3d). Western blot analysis showed the GFP-tagged truncated

gc1-a, gc1-b, gc1-c, and gc1-d with short tails could be detected at 3.0–7.6 times higher than GFP-GC1. Similarly, levels of GFP-SiGC1[104] and GFP-SiGC1[124] with short tails were 2.3–4.7 times higher than GFP-SiGC1. When these proteins were treated with cycloheximide (CHX, a protein synthesis inhibitor) for 6 h, WT long-tail Gγ proteins (GFP-GC1 and GFP-SiGC1) were rapidly degraded to undetectable levels while all of the short-tail Gγ proteins remained (Fig. 3e).

In addition to the fact that GFP-gc1-a could accumulate at 4.3-fold higher levels than GFP-GC1, we also observed the delayed degradation of GFP-gc1-a compared to GFP-GC1 when proteins were treated with the proteasome inhibitor MG132 (Fig. 3f). This revealed the existence of 26S proteasome-dependent degradation of GC1 in vivo. Therefore, gc1-a (short tail) showed greater stability than GC1 (long tail). Taken together, these results indicate that the truncated variations causing short tails of GC1 accumulate at higher protein levels and are more stable due to decreased sensitivity to C-terminus mediated proteolysis, resulting in stronger inhibition of glume coverage when compared to wild type of GC1.

### gc1-a shortens glume size by inhibiting cell proliferation. The expression pattern of GC1 varied in tissues of NIL-GC1 and NIL-gc1-a. GC1 was strongly expressed in early developing panicles (with the highest expression in the 3–6 cm young panicles), and very low expression in roots, stems, leaves, and panicles after the booting stage (Fig. 4a). At the 3 cm young panicle stage, the GC1 signal started to emerge in the whole spikelet. GC1 transcripts were localized in the pistil and stamen primordia and showed slightly lower expression in the glumes (Fig. 4b). In order to study how GC1 participates in glume development, we subsequently performed histological experiments in the NILs. Compared to NIL-GC1, from the transition to the reproductive period, NIL-gc1-a lines showed an ~35% reduction in cell number in the longitudinal innermost layer of glumes. Because there was no significant difference in cell size, the decrease in cell number resulted in a smaller glume size in NIL-gc1-a than that in NIL-GC1 (Fig. 4c, d and Supplementary Fig. 8a). A similar result was observed in the transverse section of glumes at the 1–2 cm young panicle stage (Supplementary Fig. 8b, c). Furthermore, the significantly lower glume cell number and total glume cell area in NIL-gc1-a appeared from the beginning of differentiation stage (primordia) and continued to 3–6 cm young panicle stage (Fig. 4e and Supplementary Fig. 8d). These results indicate the truncated gc1-a allele induced low glume coverage by repressing cell proliferation in the early developing panicles.

To further investigate the possible regulatory role of GC1, we performed genome-wide transcriptome profiling by RNA-seq at different developmental stages of panicles (Supplementary Data 4). Differentially expressed genes (DEGs) were mainly enriched in cell proliferation-related, transcription factors (TFs) related pathways, ATP binding, catalytic activity, and protein kinase activity pathways (Supplementary Fig. 8e and Supplementary Data 5). Total 23 TFs simultaneously appeared in 0–3 and 3–10 cm young panicles were differentially expressed between NIL-GC1 and NIL-gc1-a (Supplementary Fig. 8f). Among them, several SbMADS genes were presented but they are in different group with rice OsMADS1, which acts as a downstream effector of GS3[15]. Beyond that, total six DEGs related to Cyclin-Cyclin dependent kinase (CDK) pathway involved in cell proliferation were detected and all of them were downregulated in NIL-gc1-a compared with NIL-GC1 (Fig. 4f and Supplementary Data 6). We confirmed the transcripts of SbCYCA2;3, SbCYCB2;2 and SbCDKB1;1 were indeed significantly decreased in NIL-gc1-a compared with NIL-GC1 at the early young panicle stages (Fig. 4g). A-type cyclin

CYCA2;3 acts as a crucial regulator to terminate endoreduplication and promote cell division[16], while the B-type cyclin CYCB2;2 functions in promoting the G2/M transition with an increase in cell number in Arabidopsis[17,18]. Plant specific B-type CDK CDKB1;1 can directly accelerate mitotic cell division[19]. Thus, we posit that higher protein levels of the truncated gc1-a repress glume cell proliferation by downregulating Cyclin-CDK related gene transcripts, ultimately resulting in smaller glume size.

### A potential GC1-SbpPLAII-1 interactive module for glume coverage regulation. To further understand the GC1 working module involved in glume development, we screened for truncated gc1 interacting proteins by immunoprecipitation in combination with mass spectrometry (IP-MS). Among 1037 detected proteins, a membrane-localized patatin-like phospholipase AII-1 (SbpPLAII-1) attracted our attention (Supplementary Fig. 9a, b and Supplementary Data 7) since previous studies showed Gγ subunit 7 (GNG7) can inhibit cell proliferation in human Hela and U2OS cells[20], while human group VIA phospholipase A2 (iPLA2) generally promotes proliferation of cancer cells[21–23]. This may suggest an antagonistic relationship between Gγ and phospholipase A2 in regulating cell division. We tested the relationship between GC1 and SbpPLAII-1 by using Luciferase complementation imaging (LCI) as well as pull down and Co-immunoprecipitation (CO-IP) assays, and found that both GC1 and truncated gc1-a proteins physically interacted with SbpPLAII-1 in vivo and in vitro (Fig. 5a–c and Supplementary Fig. 10a). We additionally constructed various domain-based versions of GC1 for LCI and Bimolecular fluorescent complimentary (BiFC) assays, and observed that only the GC1-T domain was sufficient for interaction with SbpPLAII-1 (Supplementary Fig. 10b, c) while the GC1-G domain was required for interaction with SbGβ (Supplementary Fig. 10d, e). This is partly similar to interaction between GS3-1 (or GS3-4) and RGB1 in rice[11]. Furthermore, sub-cellular localization assays showed that both GFP-tagged GC1 and gc1-a, as well as SbpPLAII-1 proteins were all localized to and merged well in the cell membrane (Supplementary Fig. 11).

To confirm the effects of SbpPLAII-1 on cell proliferation, we generated millet Ci846 transgenic plants expressing SbpPLAII-1 cDNA driven by the ubiquitin promoter (Supplementary Fig. 12). At the flowering stage, all three independent SbpPLAII-1-OE plants had developed longer glumes than Ci846; and we found the longer glumes were mainly the result of increased cell numbers (Fig. 5d). Furthermore, we tested the transcripts of Cyclin-CDK related genes in the early young panicles of SbpPLAII-1-OE and Ci846. Indeed, SiCYCA2;3, SiCYCB2;2, and SiCYCB2;2, the three homologs to sorghum SbCYCA2;3, SbCYCB2;2, and SbCYCB2;2, respectively, were all upregulated in SbpPLAII-1-OE when compared with WT (Fig. 5e). These results indicate that SbpPLAII-1 acts as a positive regulator in glume cell proliferation by promoting the transcripts of Cyclin-CDK-related genes.

To determine the molecular mechanism of interaction between GC1 and SbpPLAII-1, a gradient of concentrations of GFP-GC1 or GFP-gc1-a proteins were incubated with equal amount of Flag-SbpPLAII-1. Western blot results showed that, compared to GC1, gc1-a was associated with increased degradation of SbpPLAII-1 in vivo (Fig. 5f). These results indicate that stabilized gc1-a could accelerate the degradation of SbpPLAII-1 and thus the decreased SbpPLAII-1 in plant finally affects the glume cell proliferation. However, it should be noted that we don't have double loss-of-function mutants to support the genetic interaction between GC1 and SbpPLAII-1.

### Positive selection of the truncated GC1 alleles in naked sorghum varieties. To gain genetic evidence of human selection

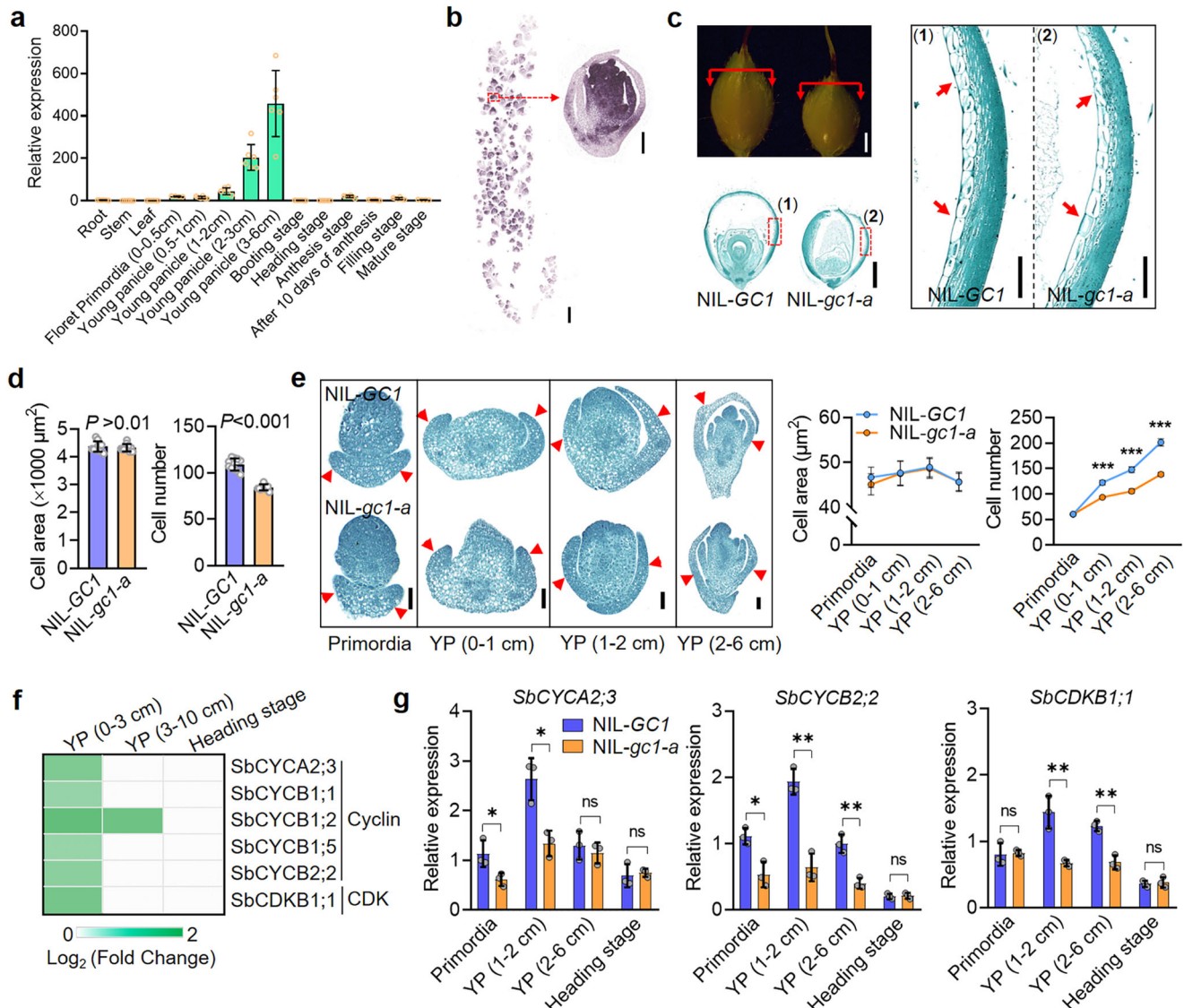

**Fig. 4 gc1-a induces low glume coverage by repressing cell proliferation. a** Spatio-temporal expression pattern of *GC1*. Data are mean ± s.e.m. *n* = 6 biological replicates. **b** RNA in situ hybridization of *GC1* in NIL-*GC1* at 3 cm young panicle stage. Left bar = 2 mm. Right bar = 200 μm. **c** Longitudinal paraffin-section of glumes in NIL-*GC1* and NIL-*gc1-a* at flowering stage. Bar = 1 mm. Magnified glume cell morphology in NIL-*GC1* and NIL-*gc1-a*. Bar = 200 μm. **d** Statistics of glume cell area and cell number in NIL-*GC1* and NIL-*gc1-a*. **e** Longitudinal paraffin-section of glumes in NIL-*GC1* and NIL-*gc1-a* at four young panicle developmental stages. YP Young panicle. Bar = 200 μm. The red arrows show corresponding glume cells. Statistical data in **d** and **e** are mean ± s.e.m. *n* = 10 biological replicates. *P*-values were determined by two-tailed unpaired *t*-test. ***Significant probability level at *P* < 0.001. **f** Heatmap of downstream DEGs of *GC1* related to Cyclin-CDK pathway. Fold change was calculated for NIL-*GC1* samples compared with NIL-*gc1-a* samples. **g** Relative gene expression of Cyclin-CDK related genes *SbCYCA2;3*, *SbCYCB2;2* and *SbCDKB1;1* at four panicle stages. Data are mean ± s.e.m. ns not significant. Three biological repeats were performed. *P*-values were determined by multiple two-tailed unpaired *t*-test. *Significant probability level at *P* < 0.05. **Significant probability level at *P* < 0.01. Source data are provided as a Source Data file.

against high glume coverage, we performed a geographic distribution analysis of haplotypes from the 482 representative genotyped sorghum accessions, which originated 38 sorghum planting countries worldwide (Supplementary Data 1). We found that 68.2% of sorghum accessions derived from 34 countries had the wild type *GC1*, which exhibited hulled and tenacious glumes. This may be a long-term adaptive and natural choice to resist predation of animals and pathogen attacks in the severe conditions by themselves[24,25]. However, the mutated *gc1-a* allele was found in 26.2% of the sorghum accessions and spread to 19 countries in Africa, America, Asia and Oceania; the other three alleles (*gc1-b/c/d*) were present in 5.6% of accessions and localized in African countries and the United States. Furthermore, we

found the four truncated alleles, *gc1-a/b/c/d*, were detected in 31.8% of sorghum accessions derived from 22 countries, suggesting these favorable haplotypes with low glume coverage were already used for local sorghum breeding of naked grains (Fig. 6).

Previous studies hypothesized that the Eastern Sahelian zone around Chad, Sudan, and northwest Ethiopia was the most likely center of sorghum domestication based on archeological evidence[26]. Through the geographic distribution of the *GC1* haplotypes, we found that most of the rare haplotypes (*gc1-b/c/d*, with each frequency in all accessions < 3%) existed in areas approximately localized to the Sahelian zone. Ten countries within the Sahelian zone contribute more than 65% of sorghum production of Africa, which may suggest a large number of

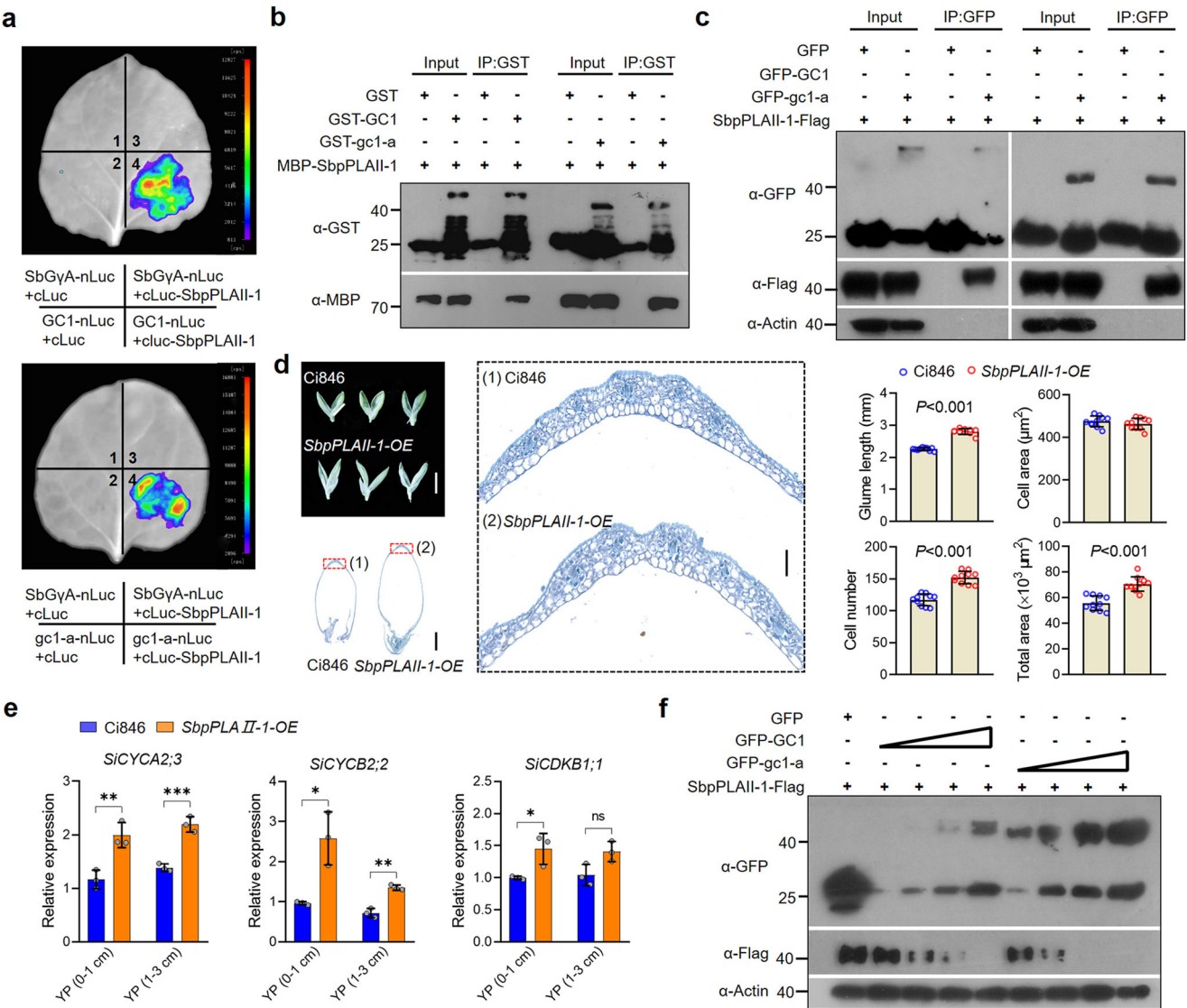

**Fig. 5 Both GC1 and gc1-a interact with and promote the degradation of SbpPLAII-1. a** GC1-nLuc and gc1-a-nLuc were co-transformed into tobacco leaves along with cLuc-tagged SbpPLAII-1 proteins by LCI assays. The typical sorghum G protein γ subunit of type A (SbGγA) and the empty cLuc were used as negative controls. Protein detection in this assay is shown in Fig. S10A. **b** In vitro pull-down assays show both GST-tagged GC1 and GST-tagged gc1-a physically interact with MBP-tagged SbpPLAII-1. The MBP-tagged SbpPLAII-1 protein pulled down with GST-GC1 or GST-gc1-a was detected by anti-MBP antibody. The combination of GST and MBP-SbpPLAII-1 was used as a negative control. **c** GFP-tagged GC1 and GFP-tagged gc1-a both interact with Flag-tagged SbpPLAII-1 in co-immunoprecipitation assays. Each protein was independently extracted from tobacco leaves. The SbpPLAII-1-Flag protein co-precipitated with GFP-GC1 or GFP-gc1-a was detected by anti-Flag antibody. The combination of GFP and SbpPLAII-1-Flag was used as a negative control. **d** Glume architecture and glume cell morphology by longitudinal paraffin-section in the wild type Ci846 and *SbpPLAII-1-OE* millet lines at flowering stage. Top left bar = 2 mm. Bottom left bar = 500 μm. Right bar = 50 μm. Statistics Data are mean ± s.e.m. n = 10 biological replicates. *P*-values were determined by two-tailed unpaired *t*-test. *Significant probability level at *P* < 0.05. **Significant probability level at *P* < 0.01. ***Significant probability level at *P* < 0.001. **e** Gene expression of Cyclin-CDK related genes in *SbpPLAII-1-OE* millet plants. Gene expression detection of *SiCYCA2;3*, *SiCYCB2;2* and *SiCYCB2;2* (the homologs to *SbCYCA2;3*, *SbCYCB2;2* and *SbCYCB2;2*, respectively) in the early young panicles of *SbpPLAII-1-OE* millet plants by qPCR assays. Three biological repeats were performed. *P*-values were determined by multiple two-tailed unpaired *t*-test. **f** GFP-tagged GC1 and gc1-a proteins can promote the degradation of Flag-tagged SbpPLAII-1. Total proteins of GFP, GFP-GC1, GFP-gc1-a, and Flag-SbpPLAII-1 were individually extracted from tobacco leaves. A gradient of concentrations (1/10, 3/10, 5/10, and 10/10 ratio of 100 μL volume) of GFP-GC1 or GFP-gc1-a protein was incubated with Flag-SbpPLAII-1 protein (10 μL volume). Each reaction was topped up by cell lysis from wild type tobacco leaves. GFP protein was used in control reaction. See in "Methods" section. Source data are provided as a Source Data file.

farmers engaged in sorghum production have selected naked sorghum varieties with rare *GC1* alleles. Among that, Nigeria shows a possible domestication area with the most variety of rare alleles since it has been the second largest sorghum producer worldwide (Fig. 6). Overall, based on the geographic distribution of the rare truncated alleles of *GC1*, we propose the Sahelian zone as a potential domestication center for naked sorghum varieties.

If truncated alleles of *GC1* were targeted by human selection of naked grains, the trace of past selection should be evident in its genomic level of nucleotide diversity (Pi). Thus, we analyzed sequence variation of *GC1* gene in three representative sorghum subgroups to identify a potential selection signal (Supplementary Data 8). The statistical results showed significantly reduced nucleotide diversity (π) and π ratio within the exon 5 and 3′UTR

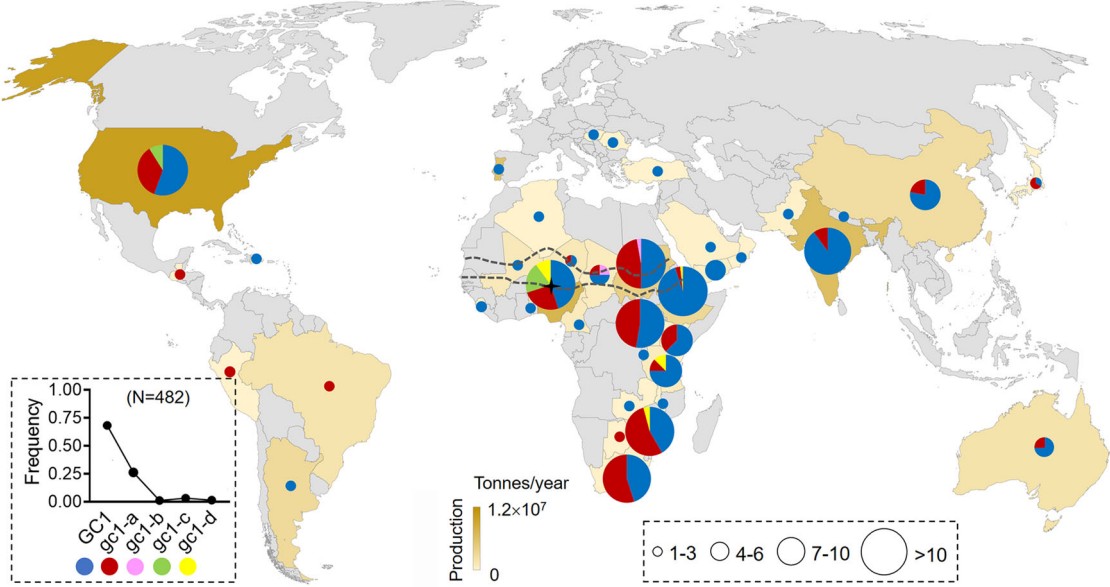

**Fig. 6 Geographic distribution and selection of *GC1* haplotypes.** Geographic distribution of the wild type *GC1* (blue color) and the other four mutated *GC1* haplotypes in 38 countries worldwide (dark yellow). The pie chart indicates the frequency of haplotypes in each country. The black dot chart shows frequency of each haplotype among the total 482 sorghum accessions. *GC1* was found in thirty-four countries while *gc1-a* occurred in nineteen countries. N number. Rare haplotypes *gc1-b*, *gc1-c*, and *gc1-d* were colored in pink, green and yellow. Countries with most of rare haplotypes were approximately localized to the Sahelian zone (gray dotted line). The black star shows Nigeria, which contains four *GC1* haplotypes simultaneously. Circle size represents the number of sorghum accessions. The color depth of each country on the map indicates the average annual production of sorghum in recent 20 years. Source data are provided as a Source Data file.

(region IV) of *GC1* in naked sorghum landrace and improved lines when compared to wild sorghum (Supplementary Fig. 13a). Moreover, a strong positive selection signature with the significantly negative Tajima's *D*-statistics and a maximum fixation index (*Fst*) value of 0.34 in the fifth exon and 3′UTR region suggests the fixed genetic differentiation between the wild sorghum and naked sorghum cultivars (Supplementary Fig. 13b). In summary, our results show that the truncated *GC1* alleles associated with naked grain morphology are a target of positive artificial selection in sorghum evolution.

## Discussion

The improvement of spikelet architecture with smaller glume size (i.e., naked grains) has been an important target in cereal cultivation and breeding systems. One of the key events in maize domestication was the elimination of hulled glumes, which was caused by an amino acid mutation in the squamosa promoter binding protein *tga1*[27]. In wheat, two reported loci (*Tg* and *Sog*) may decrease the toughness of glumes and enhance threshing rate[28,29]. Low glume coverage has been used to categorize five subtypes of domesticated sorghum (bicolor, durra, guinea, caudatum, and kafir)[30,31]. However, we observed very few naked grains among 916 millet germplasms[32], suggesting that low glume coverage may not have been a selective target during millet evolution, as their glumes are very thin and easily removed in postharvest threshing.

The heterotrimeric G-protein complex consists of Gα, Gβ, and Gγ subunits that participate in determining grain size in plants[33]. Three atypical Gγ proteins in rice, DEP1, GGC2 and GS3, depend on OsGβ to antagonistically control grain size. DEP1 and GGC2 positively increase grain length when in complex with OsGβ while GS3 reduces grain length by competitively interacting with OsGβ[11]. Two Gγ subunits, AGG1 and AGG2, can regulate various issue development and defense response by Gβγ-mediated signaling in *Arabidopsis*[34]. The interaction between SbGβ and

GC1 may also suggest a Gβγ-mediated signaling in sorghum. DEP1 and GS3 interact with the proposed downstream effector OsMADS1 to enhance its transcriptional activity, which in turn promotes the activation of target genes which are involved in the auxin synthesis and auxin transduction and affects grain length in rice[15]. Through the downstream enriched DNA binding TFs pathway of *GC1*, we found some differentially expressed genes encoding MADS, MYB, NAC, bZIP, and bHLH TFs. Some of them are possibly potential direct or indirect target effectors for GC1 and SbGβ dimers involved in floret development since the transcription levels of these genes were altered by *GC1* variations in sorghum.

In rice, the truncated version of GS3, GS3-4, has a short tail and shows shorter grain length than plants with the full version of GS3. Another truncated Gγ-like protein, dep1, has a relatively short tail and is associated with shorter grain length in rice than plants with the full DEP1 protein[11]. Recent reports showed the C-terminal tail of GS3 could be recognized and highly ubiquitinated by an E3 ligase, CLG1, while this was not the case for gs3-4[35]. This may give an explanation for the C-terminus mediated proteolysis of Gγ-like protein with long tails in crops. Some studies have provided potential models for G-proteins coupled with phospholipases working as a module in plants[36,37]. Phospholipases can transmit signals to downstream secondary molecules by lipid mediators, some of which may continue the propagation of mitogenic signals[38]. Here we identified a positive regulator, SbpPLAII-1, for glume cell proliferation involved in a phospholipase AII signaling pathway. How Gγ-like subunits promote the degradation of SbpPLAII-1 involved in cell proliferation in plants is an area for future study. A series of studies showed that phospholipase is one of known effectors downstream of G-proteins and is involved in cell division[7,8,23,39]. Additionally, we found Gγ-like subunit of GC1 and gc1-a can promote the degradation of SbpPLAII-1 at protein level. Thus we propose that the truncated gc1-a protein (short tail) is more stable than GC1 (long tail), promoting degradation of the positive regulator of cell

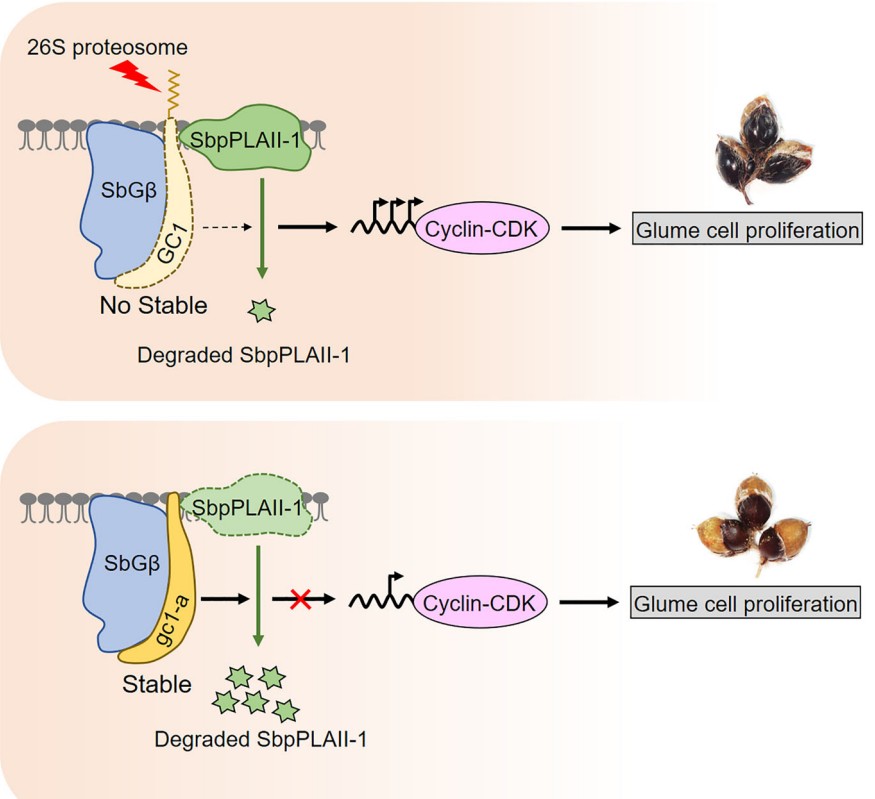

**Fig. 7 A proposed working model for GC1 regulates SbpPLAII-1 degradation and controls glume coverage.** G protein β subunit and G protein γ subunit GC1 could form a dimer to conduct a Gβγ-mediated signaling. SbpPLAII-1 functions as a positive regulator in glume cell proliferation by promoting the transcripts of Cyclin-CDK related genes. The G protein γ subunit can interact and promote the degradation of SbpPLAII-1. In wild type, the long tail of GC1 could increase the 26S proteasome-dependent degradation, resulting in an unstable protein level. This subsequently leads to a normal phospholipase SbpPLAII-1 level and results in a final high glume coverage phenotype in sorghum. However, the truncated gc1-a protein (short tail) is more stable than GC1 (long tail) due to the absence of C-terminus, and then enhances the degradation of SbpPLAII-1. It leads to a significantly inhibited SbpPLAII-1 signal involved in glume cell proliferation and therefore causes low glume coverage in sorghum.

division SbpPLAII-1, ultimately resulting in inhibition of cell proliferation and therefore causing low glume coverage in sorghum (Fig. 7). Additional genetic evidence could reveal the relationship between GC1 and SbpPLAII-1.

The effects of Gγ-like proteins on yield-related traits may differ among crops. Rice *GS3* is a major gene for grain size and can explain 35–79% of phenotypic variation for grain length in 180 rice variaties[12,40]. However, the phenotypic variation for thousand kernel weight given by maize *ZmGS3* is less than 8%[41], and grain size variation contributed by sorghum *GC1* is only 4–10%[42]. It suggests that both *ZmGS3* and *GC1* are minor genes for variations in grain yield. The other possibility could be due to the degeneration of rice glume (hull) and the grain size is mainly controlled by lemma and palea, which is different to maize and sorghum. Although *GC1* appears to act as a negative regulator of grain size in a segregated population[42], we did not find a high correlation signal between *GC1* and yield-related traits in the diverse sorghum panel (Fig. 1e and Supplementary Fig. 14a), indicating phenotypic variation of yield-related traits explained by *GC1* is not high in the natural sorghum accessions due to its minor effects on grain size. Moreover, we found certain portion of varieties with naked but large grain size in sorghum germplasms (Supplementary Fig. 14b). It may suggest that the slight loss of grain yield caused by truncated gc1-a can be overcome by combining other major loci, such as *qGW1a* that contributes 20–40% grain weight in sorghum[43]. In summary, de novo domestication by modern technology such as gene-editing could accelerate

future sorghum breeding based on our discovery of *GC1* alleles associated with glume coverage variation.

## Methods

**Materials**. A diverse sorghum panel including 915 accessions was collected and used to evaluate the morphological degrees of glume coverage and glume length. For GWAS, we used the Sorghum Association Panel (SAP) consisting of 352 lines[10]. The SAP population was planted under completely random block design in three cities in China (Beijing [39.5°N, 116.4°E], Yinchuan [38.5°N, 106.3°E] and Sanya [18.3°N, 108.8°E]), with two replicates in each planting site in 2016 and 2017. The seeds of the natural SAP population were obtained from the United States Department of Agriculture-Agricultural Research Service (USDA-ARS).

For the mapping population, we developed an F6 generation including 188 individuals derived from a cross between two parental lines, SN010 (hulled) and M-81E (naked). SN010 is a local sweet sorghum landrace from Shanghai, China, while M-81E is a sweet sorghum improved line from the United States. The initial *GC1* interval was flanked by two SSR markers: *MSR0061* and *MSR0084*. The other SSR marker, *SAM39569*, was co-segregated with *GC1* in the F6 population. To fine-map the *GC1* locus, three independent residual heterozygous lines (RHLs), RHL-96/97/101, were identified with a heterozygous genotype in the initial *GC1* region and maximum homozygous genotypes in remaining backgrounds from the F6 generation. All three RHLs were self-crossed into the next two enlarged populations: the F7 population (1084 plants) and the F8 population (594 plants). The F8 population was generated from RHL97-55 of the F7 population. Forty recombinants of four types in the *GC1* locus were screened by designed markers. One randomly selected plant with heterozygous genotype in the *GC1* locus from F8 population was self-crossed two times into F10 generation. NILs were identified by flanking markers *MSR0069* and *MSN0013* in the F10 generation. The parental lines and above populations were planted in the rows were at intervals of 50 cm and the columns were at intervals of 20 cm. The field managements of watering, weeding and fertilizing were followed by the local cultivated conditions. Primers are shown in Supplementary Data 9.

**Phenotype evaluation.** Five to ten fresh grains with glumes of every panicle were checked. Glume length and glume width were calculated in average data. Glume coverage was scored in the SAP population by five levels: 1 (Very low glume coverage), 2 (Low glume coverage), 3 (Moderate glume coverage), 4 (High glume coverage), and 5 (Very high glume coverage). Final glume coverage data in the natural SAP population that could repeat at least eight times out of 12 planting repeats were accepted, otherwise were evaluated as missing data. Glume coverage in mapping populations was identified as a quality trait, which scored as 1 (Low glume coverage), 2 (Moderate glume coverage), and 3 (High glume coverage).

**GWAS and markers development.** Reference genome sequences of BTx623 and annotation files were downloaded from the Phytozome website (https://phytozome.jgi.doe.gov)[44]. Raw reads of resequencing sorghum lines from the SAP population were publicly available from the community resource[10] and were filtered by Illumina's quality control filter and adapters removed. Clean reads were aligned to the reference genome using the bwa-mem method in BWA (v0.7.16)[45]. SNPs and indels of each line were called by SelectVariants and were filtered by VariantFiltration of using GATK with the following parameters: QD < 2.0, FS > 200.0, ReadPosRankSum < −20.0[46]. GWAS analysis was performed with 82,430 SNP markers[47] using a compressed mixed linear model (cMLM). The significant P-value threshold was determined by Bonferroni correction[48]. For initial mapping, the major locus for glume coverage was investigated with a total of 53 reported SSR markers[49] and 2 SSR and 18 indel designed markers with polymorphism among the two parents; Single marker analysis was conducted with the Icimapping[50]. One SSR, 2 indel and 2 SNP markers were designed and screened for fine mapping. All primers used in this study are shown in Supplementary Data 9.

**qPCR and RNA-Seq.** Total RNA was extracted from sorghum and millet panicles at different developmental stages using pure RNA extraction kit (Huayueyang, Beijing, China), with DNA removed by DNase I treatment. Reverse transcription of each sample was carried out with a cDNA synthesis kit (TransGen, Beijing, China). qPCR was performed using SYBR Green mix (TransGen, Beijing, China) on an Applied Biosystems 7900HT Fast Real-Time PCR System. Quantitative variations were calculated by the relative quantification method ($2^{-\triangle\triangle CT}$ [DDCT]) and three to six independent repeated experiments were performed for each sample. Primer sequences are shown in the Supplementary Data 9.

Total RNA of fresh samples of NIL-GC1 and NIL-gc1-a at 0–3 cm and 3–10 cm young panicles, and heading stages were extracted and used to perform RNA-Seq. Each sample was comprised with three independent plant tissues. The significant differentially expressed genes (DEGs) shown in Supplementary Data 4 were determined by the cutoff of Log$_2$ (fold change) (GC1 vs. gc1-a) value ≥1 and adjusted P-value < 0.001. The downstream DEGs were analyzed for enriched pathways by Gene Ontology[51] and Kyoto Encyclopedia of Genes and Genomes (GO-KEGG)[52]. See details in Supplementary Data 5.

**Transgenic overexpression and genome editing.** For GC1 and gc1 overexpression in sorghum, the full-length cDNAs of GC1 and gc1 were each cloned into the pCambia2300-Myc vector with a Myc tag driven by ubiquitin promoter. The SiGC1[124] and SbpPLAII-1 overexpression constructs were generated in the same way. GC1-OE and gc1-OE were introduced into the recipient sorghum line Wheatland, which contains the wild type GC1 allele, using Agrobacterium tumefaciens EHA105. SiGC1[124] overexpression and heterologous expression of SbpPLAII-1 in the recipient millet line Ci846 were carried out by efficient Agrobacterium-mediated transformation[53]. The positive transgenic overexpressing plants were confirmed by PCR and the gene expression level of target genes in the T$_0$ generation were verified by qPCR.

The various domain-based versions of GC1 and SiGC1 mutants were generated through genome editing by CRISPR/Cas9 technology[13] to verify the complex function of genes in sorghum and millet. The single mutant GC1-KO was generated using the sgRNA target designed in the second exon of GC1 driven by the rice OsU6a promoter. The single mutant SiGC1-KO was generated with the sgRNA target designed in the first exon of SiGC1 driven by the rice OsU6a promoter. The genome editing experimental steps are as follows.

A 20 bp specific sgRNA target sequence of GC1 and SiGC1 was selected. The forward and reverse target sequences were synthesized and annealed to form the oligonucleotide adapters with double chain. The adapters were each connected to the pYLgRNA-OsU6a vector. The purified gRNA expression cassettes were inserted into the pYLCRISPR-Cas9-MB/MH vectors (MB for sorghum and MH for millet). The ultimate recombinant CRISPR-Cas9 plasmid was introduced into the recipient sorghum line Wheatland and the recipient millet line Ci846 through Agrobacterium-mediated transformation. The heterozygous mutations within the targets of each mutant in the T$_0$ generation were amplified by PCR using primers flanking the target sites. The homozygous mutants were identified in the target sites in the T$_1$ generation. Primers are shown in Supplementary Data 9.

**Histology and RNA in situ hybridization.** Fresh young panicles of NIL-GC1 and NIL-gc1-a were fixed in FAA solution (50% ethanol, 5% acetic acid, and 3.7% formaldehyde), followed by vacuuming for 20 min. The tissues were treated with gradient concentration of ethanol and xylene for dehydration and infiltration, and

subsequently were soaked with paraffin. After embedding in paraffin, the paraffin blocks were sliced into 8 μm longitudinal or transverse sections by microtome (LEICA RM2016). Slices were subsequently dyed using Safranin O-Fast Green staining (SERVICEBIO G1031) or Toluidine Blue O staining (CAS 3209-30-1). A 27 bp fragment across the fourth and fifth exon in the cDNA region of GC1 was designed and used as the template to make the antisense RNA probe. We used the DIG DNA Labeling and Detection Kit (CAS 11093657910) to generate the antisense RNA probe with digoxigenin-label. RNA in situ hybridization with the antisense probe was performed on longitudinal sections of young panicles. Final cell morphology of each slide was observed through a microscope (NIKON ECLIPSE E100) and scanned with an imaging system (NIKON DS-U3).

**Protein detection by western blot.** The cDNA of various domain-based versions of GC1 and SiGC1 alleles was amplified and inserted into the pCambia1300-GFP vector. Each construct was transformed into A. tumefaciens strain GV3101. The Agrobacterium strain was independently infiltrated into N. benthamiana leaves with p19. Leaves were collected after three days. Total protein was extracted with the cell lysis buffer (50 mM Tris-HCL, 50 mM sucrose, 150 mM NaCl, 0.1% Triton X-100, 1 mM EDTA, 0.2% NP-40, and 1 mM protease inhibitor PMSF). Plant extracts were separated by SDS-PAGE electrophoresis. The proteins were transferred to a nitrocellulose filter membrane under 100 volts for 75 min. The membrane was blocked in 5% nonfat milk for 1 h and subsequently immunoblotted with anti-GFP antibody (Novoprotein AB006-01A) for 1 h. The membrane was washed in PBST (20 mM Tris, 150 mM NaCl, 0.1% Tween 20) and then incubated with Goat anti-mouse secondary antibody (Proteintech SA00001-1) for 1 h. All original uncropped immunoblot images are shown in Source Data.

**Protein degradation assay.** The coding sequence of GC1 and gc1-a were cloned into the C-terminal of GFP tag fusion pCambia1300-GFP vector while the cDNA of SbpPLAII-1 was cloned into the N-terminal of Flag tag fusion pCambia1300-221-Flag vector. The Agrobacterium strains containing 35S::GFP-GC1 (OD = 1.5), 35S::GFP-gc1-a (OD = 1.5) and 35S::SbpPLAII-1-Flag (OD = 1.5) were independently infiltrated into tobacco leaves with p19 (OD = 1.0). The infiltrated leaves were used to extract total protein of each construct with the above cell lysis buffer (50 mM Tris-HCL, 50 mM sucrose, 150 mM NaCl, 0.1% Triton X-100, 1 mM EDTA, 0.2% NP-40, 1 mM protease inhibitor PMSF, 50 μM MG132, and 10 mM ATP).

A gradient of concentrations of GFP-GC1 or GFP-gc1-a protein (10, 30, 50, and 100 μL volume) was incubated with equal SbpPLAII-1-Flag protein (10 μL volume). GFP protein was used in a control reaction. Each reaction volume was topped up by total protein extracted from empty tobacco leaves. Reactions were placed in a 200-rpm shaker for 6 h at 28 °C. Protein detection was performed by western blot as described above. The membrane was finally immunoblotted with anti-GFP antibody and anti-Flag antibody (Proteintech 20543-1-AP). Plant actin was used as the loading control. Three biological repeats were performed.

**IP-MS.** The truncated gc1 interacting protein screen was performed by immunoprecipitation in combination with mass spectrometry (IP-MS). Total protein was extracted from the mixed 2–3 cm young panicles of three independent gc1-OE transgenic plants. The extracted protein was precipitated by anti-Myc magnetic agarose (MBL, M047-8) and purified with washing buffer (10 mM Tris-HCl, 150 mM NaCl, 0.5 mM EDTA, 0.5% NP-40, 1 mM protease inhibitor PMSF and 1× protease inhibitor cocktail) at least five times. The beads were added to a 1× loading buffer and boiled for 20 min, then samples separated through SDS-PAGE gel electrophoresis. After in-gel digestion, the samples were used for mass spectrometry (Thermo Scientific, nanoLC-Q EXACTIVE). Information about interacting proteins of gc1 by IP-MS was shown in Supplementary Data 7.

**LCI and BiFC assays.** The cDNA of SbGα (Sobic.001G484200), SbGβ (Sobic.001G142100), various domain-based versions of GC1 alleles, gc1-a and SbpPLAII-1 were amplified and inserted into the N-terminal of Luc fusion vector pCambia1300-nLuc and the C-terminal of Luc fusion vector pCambia1300-cLuc for the LCI assays[54]. Transient expression in tobacco leaves was conducted as described above. After 3 days, 1 mM luciferin (Promega, E1605) was applied to the injected regions of the tobacco leaves. The Luc fluorescence signal was observed with CCD-imaging instrument (Berthold, LB985). The expressed nLuc-tagged and cLuc-tagged proteins in tobacco leaves were detected using anti-Luciferase antibody (Sigma, L0159). Plant actin was used as a loading control.

Similarly, these clones were also inserted into the pSPYNE(R)173 and the pSPYCE(R) for BiFC assays[55]. The fluorescence was captured with total internal reflection fluorescence microscopy (TIRF3, Axioimager.Z2). Relevant primer sequences are shown in Supplementary Data 9.

**In vitro pull-down assays.** The coding sequences of GC1 and gc1-a were cloned into the C terminal of GST tag fusion pGEX-6p-1 vector. The cDNA of SbpPLAII-1 was cloned into the C-terminal of MBP tag fusion pMal-c2X vector. Each of purified recombinant bait protein GST, GST-GC1 and GST-gc1-a, independently along with MBP-SbpPLAII-1 were both added to the binding buffer (50 mM Tris-HCl, 100 mM NaCl, 0.6% Triton X-100, 0.2% glycerol, and 1 mM PMSF). The

mixture was incubated overnight at 4 °C. Glutathione Sepharose 4B beads (GE, 17-0756-01) were added. Samples were incubated for 2 h at 4 °C, and subsequently washed with the binding buffer at least five times. The pulled-down proteins by Glutathione Sepharose 4B beads were detected by immunoblotting with anti-MBP antibody (Sigma, M6295).

**Co-immunoprecipitation**. Total protein from the infiltrated tobacco leaves for each construct was independently extracted as described above. After GFP, GFP-GC1 and GFP-gc1-a independently incubated with SbpPLAII-1-Flag in the cell lysis buffer (50 mM Tris-HCL, 50 mM sucrose, 150 mM NaCl, 0.1% Triton X-100, 1 mM EDTA, 0.2% NP-40, 1 mM protease inhibitor PMSF, and 50 μM MG132) for 2 h at 4 °C, the protein precipitated by GFP-trap magnetic agarose (Chromotec, gtma-20) was detected by western blot with anti-Flag antibody. Plant actin was used as the loading control.

**Subcellular localization**. Subcellular localization of GFP-tagged GC1, gc1-a and GC1-T were performed using the pCambia1300 construct of the coding sequence of *GC1* or *gc1-a* allele with a GFP tag driven by a *CaMV35S* promoter. The same vector backbone with a substituted mCherry tag was used to generate the mCherry-tagged SbpPLAII-1 construct. All of the fusion constructs were transformed into *Agrobacterium tumefaciens* strain GV3101, and the bacterial suspension (OD600 = 1.5) was infiltrated *into N. benthamiana* leaves with the equal proportion of *Agrobacterium* carrying p19 (OD600 = 1.0). The fluorescence signals were examined after 2 days by a confocal laser scanning microscope (LEICA TCS SP5). The empty GFP and empty mCherry constructs were used as controls. Three biological repeats were performed and the similar localization results were obtained.

**Selection analysis**. Nineteen wild sorghum accessions, twenty-five naked landrace varieties and sixty-nine naked improved lines (Supplementary Data 8) were used to perform the selection related analysis. Nucleotide diversity ($\pi$ value) and was calculated by VCFtools[56]. Tajima's $D$-statistic tests, $Fst$ values, and HKA tests around the *GC1* gene were calculated by VCFtools and DnaSP v 5.0[57].

**Reporting summary**. Further information on research design is available in the Nature Research Reporting Summary linked to this article.

## Data availability

The raw transcriptome sequence data reported in this paper have been deposited in the Genome Sequence Archive in National Genomics Data Center, China National Center for Bioinformation, Chinese Academy of Sciences, under accession number PRJCA006306. Source data are provided with this paper.

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

## Acknowledgements

We thank the Department of Agriculture-Agricultural Research Service (USDA-ARS U.S.) for providing sorghum seeds of the Sorghum Association Panel (SAP). This research was supported by grants from the Strategic Priority Research Program of the Chinese Academy of Sciences (XDA24010306), from the National Natural Science Foundation of China (U1906204), the National Key R&D Program of China (2019YFD1002701 and 2018YFD1000704), and the Agricultural Breeding Program in NingXia Province (2019NYYZ04 and 2019BBF02022-05). S.T. is supported by Youth Innovation Promotion Association of Chinese Academy of Sciences.

## Author contributions

Q.X., Y.W., and P.X. designed the project. P.X. performed most of the experiments. P.X., S.T., and C.L. constructed multiple artificial populations. P.X., S.T., and C.L. performed phenotyping traits in the field. C.C. performed the genotyping data for GWAS analysis. P.X., H.Z., Y.S., and C.W. performed the transgenic experiments in sorghum and millet. X.D. provided the millet accessions. P.X. and C.C. performed the selection analysis. P.X., C.C., Y.W., Q.X., F.Y., and H.W. analyzed the data. P.X., Q.X., and Y.W. wrote the manuscript. All authors have read, discussed and contributed to the manuscript.

## Competing interests

The authors declare no competing interests.
