## [Peer Review File · Nature Communications]

Natural variation in GC1 (Glume Coverage 1) causes naked grains in sorghumReviewers' Comments:

Reviewer #1:

Remarks to the Author:

GC1 is a fascinating story!

The reported work is complete. The GC1 was discovered by both GWAS and map-based cloning. The naturally truncated GC1 was demonstrated to confer low glume coverage by multitudes of experiments.

Compared with the interesting story and the beautiful work, the English needs some improvement. I have attached a edited copy of the manuscript. Please feel free to use or reject them. Also, I didn't have time to edit the entire manuscript. Please have the manuscript carefully edited to match the good work.

A minor point for discussion why does GS3 control rice grain size but no correlation grain size in sorghum? This point needs a more detailed discussion, even it may be speculative.

Reviewer #2:

Remarks to the Author:

This paper identified a Gy subunit GC1 in sorghum causing naked grains by determination of glume coverage. This work attracts me in three aspects. First of all, as an ortholog of GS3 in rice, GC1 in sorghum and SiCG1 in millet maintained a conserved negative regulation in organ size. However, they evolved a discrepant role in glume development but not grain size regulation possibly due to divergent target of domestication process. Second, a novel downstream network was identified connecting G-protein pathway and cell division. And natural variations of GC1 in sorghum and millet would facilitate grain threshing and thus reduce the cost of large-scale processing. The work is generally interesting, but there still exist a few questions as follows:

Major:

1. Line 140-141, the description that "The conflicting phenotypes between GC1-KO and the natural truncated versions of GC1 is an unconventional finding" is not accurate, and the authors misunderstand the function of different domains of GC1.

The Gy-like domain and the C-terminal domain have different functions. The natural truncated versions of GC1 still contain a Gy-like domain, whereas the function of GC1-KO is completely disrupted. What is the expect phenotype if the sgRNA target sites are located on the C-terminal domain? The authors should clearly define different alleles of GC1 and describe the function of each domain of GC1 to fully understand its function.

2. Line 198-201, The expression pattern of GC1 varied by tissue in NIL-GC1 and NIL-gc1-a. However, as mentioned in Fig. 3c, there were no significant differences in gene expression among the five GC1 alleles. How to understand these two results?

3. Line 216, Any other differentially expressed genes or enriched items that worth mentioning in the transcriptome data?

4. Line 247, the authors indicate that the GC1-G domain was required for interaction with G β . Does this mean that the function of GC1 is dependent on G β ?

5. Line 251, What is the expression level of Cyclin-CDK related gene in SbpPLA II -1-OE plants? How about the interaction of GC1 and SbpPLA II -1 at the genetic level and may discuss about it.

6. Any other phenotypic effect of GC1 in agronomic traits?

7. The working model in Fig. 5g might be improved to integrate more comprehensive information identified in this paper.

Minor:

1. Line 50-52, where does the information of sorghum and maize G proteins come from? The listed reference is about the rice G proteins.

2. Line 64-68, this part should be in introduction rather than in results.
3. The figure legend is not clear enough in a number of figures. For example, in Fig. S1a, the abbreviations are not annotated in the legend. In Fig. 4g, the color is not explained in the legend.
4. Line 79-80, How about the effects of other two QTLs?
5. There exist different alleles of GC1 with discrepant functions. Therefore, the authors should clearly indicate the exact form of GC1 involved in all experiments of the result part to avoid any confusion.

Reviewer #3:

Remarks to the Author:

The manuscript by Xie et al. describes the positional cloning and characterization of a major locus regulating reduced glume length present in some domesticated sorghum lines. GWAS identified two major loci controlling glume coverage, and the locus on chromosome 1 (GC1) was fine mapped to a 58 kb region containing 5 genes. A combination of expression analysis and SNP/INDEL association analysis across this region identified a candidate insertion resulting in a frameshift and truncated C-terminal mutation in an atypical G-gamma subunit homologous to a known QTL regulating rice seed size. Multiple independent frame-shift alleles causing a similar deletion in the C-terminal domain all associate with reduced glume length in diverse sorghum accessions. This deletion variant was confirmed by transgenic analysis in both sorghum and millet, which demonstrated that the deletion variant can reduce glume size and is resistant to protein degradation. Interestingly, the C-terminal deletion is likely a gain of function mutation as knock out CRISPR lines had longer glumes. Gc1 lines with decreased glume size had fewer cells, although the cell size was unaffected suggesting that GC1 affects glume length by inhibiting cell proliferation. Mass-spec analysis identified a patatin-like phospholipase A2 (sbpPLAII-1) protein interactor with GC1. Transient co-expression in tobacco indicate that GC1 accumulation leads to degradation of sbpPLAII-1, and over-expression of sbpPLAII-1 in millet increased glume length, suggesting a possible functional interaction as well. Finally, the authors examined the distribution of GC1 alleles as well as GC1 nucleotide diversity across wild vs. naked grain landraces.

Overall this is an impressive study, presenting a significant amount of work establishing a strong case that C-terminal deletion of GC1 leads to increased protein accumulation and reduced glume length. Although this gene has been identified as a QTL previously in rice, this work shows that an atypical G-gamma influences glume size as well as grain size. In addition, the interaction with a phospholipase provides a novel potential signaling mechanism to follow-up in plant G-protein signaling. While I am enthusiastic about the results, I have a few concerns regarding presentation of the data and its interpretation that should be addressed in a revised manuscript.

Major Concerns:

1. The expression levels of GC1 vs gc1-a are not consistent from figure 3c to 4a. In 3c it appears that both alleles are expressed at similar levels, while in figure 4a it appears that gc1-a is significantly reduced in expression compared to GC1. Which is correct? The other results are convincing that the C-terminal deletion and its effects on protein stability are causative. However, if the alleles have different expression levels that could indicate a change in expression pattern related to additional polymorphisms in the regulatory DNA.
2. I was confused by the presentation of the GC1 deletion lines in supplementary figure 10c. Based on these results, it looks to me like the N, G and C domains are required for interaction, since their deletion abolishes BiFC. However, the T-deletion construct maintains a positive BiFC suggesting that it does not contribute to interaction with PLAII-1. However in the text it says "only the GC1-T domain was sufficient for interaction with SbpPLAII-1" (line 246). I'm not sure what you are trying to express in the text. Are you saying the T domain is sufficient for interaction? If so, that is not consistent with the data. Rather the T-domain was the only domain that did not contribute to interaction.

4. Figure 5f is critical to the interpretation of the functional interaction of GC1 with sbpPLAII. However, the details of how this experiment was performed were unclear. I assume that these were the results of a co-inoculation of tobacco leaves? If so, how can you reliably modulate GC1 levels while keeping PLAII-1 levels constant? Was this done by altering the relative concentrations of each Agro construct in the inoculation? If so, that seems problematic since expression of each construct is variable across the tobacco cells, and you cannot assess degradation unless PLAII-1 is kept constant.

5. The nucleotide diversity analyses in figures 6b-c show trends consistent with selection. However, there is no statistical analysis of the significance of these findings. Absent this statistical analysis, it is hard to know how to interpret the results.

Minor concerns

1. Overall the manuscript is clear and easy to read, however some of the sentences are awkward, and the whole manuscript would benefit by careful editing of a native English speaker.

2. line 85: supplemental figure 2b does not show co-linearity of the M-81E allele with the GWAS results presented in figure 1b. What are the positions of the closest markers in each of these maps?

3. line 103: a C>A SNP creating a stop codon is not a frameshift, please modify the text to clarify the distinct nature of the gc1-c allele.

4. The gc1-a allele appears to be a gain of function. If so, one would predict that it is dominant or semi-dominant over GC1. Do you have measurements of a NIL-gc1-1/NIL-GC1 het that would confirm this?

5. Lines 216-229: A genome wide transcript profiling experiment was performed, but the results for only a small set of cyclin and a CDK were reported. Is there a good justification for only focusing on these genes, and ignoring all the remaining DEGs? I was surprised that this wasn't directly addressed in the text. Perhaps a supplemental figure containing a broader summary of the RNA-seq would be in order.

6. In figure 1c the N values are not given for the SN010 and M-81E lines.

7. Figure 2 e, f, and H there are no P values given to assess the significance of the different lines.

8. Figure 4g the blue vs. orange bars are not labelled.

9. Figure 6b there is no scale on the X-axis.

10. Supp figure 1a: VHGC HGC MGC LGC and VLGC are not defined.

REVIEWER COMMENTS

Reviewer #1 (Remarks to the Author):

GC1 is a fascinating story!

The reported work is complete. The GC1 was discovered by both GWAS and map-based cloning. The naturally truncated GC1 was demonstrated to confer low glum coverage by multitudes of experiments.

Compared with the interesting story and the beautiful work, the English needs some improvement. I have attached a edited copy of the manuscript. Please feel free to use or reject them. Also, I didn't have time to edit the entire manuscript. Please have the manuscript carefully edited to match the good work.

> We are very appreciated for the positive evaluation for our manuscript and the edited version. We have revised the English language and also added discussion accordingly.

A minor point for discussion why does GS3 control rice grain size but no correlation grain size in sorghum? This point needs a more detailed discussion, even it may be speculative.

> We added the following paragraph to strength this point in the discussion section.

Rice *GS3* is a major gene for grain size and can explain 35-79% of phenotypic variation for grain length in 180 rice varieties (1,2). However, the phenotypic variation for thousand kernel weight given by maize *ZmGS3* is less than 8% (3), and grain size variation contributed by sorghum *GCI* is only 4-10% (4). It suggests that both *ZmGS3* and *GCI* are minor genes for variations in grain yield. The other possibility could be due to the degeneration of rice glumes (hulls); and rice seed size is mainly controlled by lemma and palea, which is different to maize and sorghum floret.

Discussion based on the following references:

- (1) Fan et al., *Theor. Appl. Genet.* 112, 1164-1171 (2006).
- (2) Fan et al., *Theor. Appl. Genet.* 118, 465-472 (2009).
- (3) Li et al., *Theor. Appl. Genet.* 120, 753–763 (2010).
- (4) Zou et al., *J. Exp. Bot.* 71, 5389-5401 (2020).

Reviewer #2 (Remarks to the Author):

This paper identified a G γ subunit GC1 in sorghum causing naked grains by determination of glume coverage. This work attracts me in three aspects. First of all,

as an ortholog of GS3 in rice, GC1 in sorghum and SiCG1 in millet maintained a conserved negative regulation in organ size. However, they evolved a discrepant role in glume development but not grain size regulation possibly due to divergent target of domestication process. Second, a novel downstream network was identified connecting G-protein pathway and cell division. And natural variations of GC1 in sorghum and millet would facilitate grain threshing and thus reduce the cost of large-scale processing. The work is generally interesting, but there still exist a few questions as follows:

> We are very appreciated for positive evaluation for our manuscript. We have revised our MS accordingly.

Major:

1. Line 140-141, the description that “The conflicting phenotypes between GC1-KO and the natural truncated versions of GC1 is an unconventional finding” is not accurate, and the authors misunderstand the function of different domains of GC1. The G γ -like domain and the C-terminal domain have different functions. The natural truncated versions of GC1 still contain a G γ -like domain, whereas the function of GC1-KO is completely disrupted. What is the expect phenotype if the sgRNA target sites are located on the C-terminal domain? The authors should clearly define different alleles of GC1 and describe the function of each domain of GC1 to fully understand its function.

> Thanks for your suggestions. To avoid the confusion, we have modified the description by replace that sentence with “The *GCI-KO* shows high glume coverage, which is different to what we found in *gcl-a*” (Line 145-146).

We expect that the plant with sgRNA target sites located on the C-terminal domain should have similar phenotype to *gcl-a* if the edited versions could produce similar truncated proteins.

2. Line 198-201, The expression pattern of GC1 varied by tissue in NIL-GC1 and NIL-*gcl-a*. However, as mentioned in Fig. 3c, there were no significant differences in gene expression among the five GC1 alleles. How to understand these two results?

> The varied transcriptional levels in the NILs might indicate potential variation in the promoter region of the two natural alleles. We then detected sequence variation in the promoter and found a 341 bp insertion (*inser341*, see following figure a and b) located at 5'UTR of *gcl-a* allele when compared to *GCI*, despite of other multiple SNPs and small indels. Next, we designed a molecular marker flanking *inser341* to verify promoter variation in the selected sorghum accessions used for Fig. 3c, and found both *inser341* variety and gene expression level of *GCI* were not significantly correlated with the five natural *GCI* alleles, as well as glume coverage phenotype (See following figure c).

Second, there is a large range of *GCI* transcript levels (from 0.29 to 5.54 times,

see details in Source data Fig. 3c) among 10 sorghum accessions (used in Fig. 3c) with *GCI* haplotype, suggesting that there exists distinct *GCI* expression level in different sorghum accessions.

Thus, with these two evidences, we could conclude both *in*ser341 variety and gene expression level of *GCI* were not related to glume coverage phenotype. To avoid the misunderstanding between Fig. 3c and Fig. 4a, we removed the transcription level of *gc1-a* allele in Fig. 4a. That has no impact for overall results.

3. Line 216, Any other differentially expressed genes or enriched items that worth mentioning in the transcriptome data?

> Truly besides those genes in cell cycle and cell proliferation related pathways, we also found numbers of differentially expressed genes (DEGs) enriched, such as transcription factors (TFs), ATP binding, catalytic activity and protein kinase activity pathways. Among them, several *SbMADS* genes were presented but are at different group of rice *OsMADS1*, which acts as a downstream effector of *GS3*. Thus, we did not analyze them in detail but focused on those cell cycle and cell proliferation related pathways due to our physiological data on the cell division. We added a Supplementary Figure 8e and f and a short paragraph in result part (Line 226-233), as well as a discussion about the downstream TFs of *GCI* in the discussion section (Line 348-353).

4. Line 247, the authors indicate that the GC1-G domain was required for interaction with Gβ. Does this mean that the function of GC1 is dependent on Gβ?

> G β subunit and G γ subunit always interact to form a dimer to regulate plant organ development and defense response (Ford et al., Science. 280, 1271-1274 (1998) and Trusov et al., Plant Cell. 19, 1235-1250 (2007)). Sun et al. demonstrated that the three atypical G γ subunits in rice, OsGS3, OsDEP1 and OsGGC2, were functionally dependent on the OsG β (OsRGB1) (Nat. Commun. 9:851 (2018)). Since GC1-G domain interacts with G β , which is similar to what found in rice. Thus, we discussed the possibility that GC1 is dependent on G β in sorghum (Line 342-345).

5. Line 251, What is the expression level of Cyclin-CDK related gene in *SbpPLA II-1*-OE plants? How about the interaction of GC1 and *SbpPLA II-1* at the genetic level and may discuss about it.

> Thanks for your suggestions. We detected the transcripts of *SiCYCA2;3*, *SiCYCB2;2* and *SiCYCB2;2* (the homologues to *SbCYCA2;3*, *SbCYCB2;2* and *SbCYCB2;2*, respectively) in the young panicles of *SbpPLA II-1*-OE millet plants by qPCR assay (Supplementary Fig. 12c), and indeed found *SbpPLA II-1* could upregulate transcripts of these Cyclin-CDK related genes and promote glume cell proliferation (Line 271-275).

We are apologized that we have no suitable genetic materials to confirm the genetic interaction between *GCI* and *SbpPLA II-1*. So following your suggestion we discussed it in the discussion (Line 367-372) as: A series of studies showed that phospholipase is one of known effectors downstream of G-proteins and is involved in cell division (1,2,3,4). Additionally, we found G γ -like subunit of GC1 and *gc1-a* can promote the degradation of *SbpPLA II-1* at protein level. Together it may suggest *SbpPLA II-1* acts together with GC1. Additional genetic evidence could reveal the interaction between GC1 and *SbpPLA II-1*.

Discussion based on following references:

- (1) Pandey et al., Annu. Rev. Plant Biol. 70, 213-238 (2019).
- (2) Burch et al., Mol. Neurobiol. 3, 155-171 (1989).
- (3) Ma et al., Lipids 36, 689-700 (2001).
- (4) Murakami et al., BBA-Mol. Cell Biol. L. 1864, 763-765 (2019).

6. Any other phenotypic effect of GC1 in agronomic traits?

> *NIL-gc1-a* also shows the lower plant height, a slight shorter grain size and a 7-day delayed flowering time than *NIL-GCI*. *GCI*-KO has larger grain size but no significant difference in plant height when compared with WT. It suggests *GCI* may also function in plant architecture and yield-related traits.

7. The working model in Fig. 5g might be improved to integrate more comprehensive information identified in this paper.

> Thanks for your suggestion. According to previous reports and our results, we add SbGβ interacted with GC1 and Cyclin-CDK-related genes located downstream of SbpPLA II-1 in Fig. 5f. This will better describe a novel G-protein pathway that truncated gc1-a inhibits cell proliferation through a SbpPLA II-1 signal, resulting in low glume coverage of sorghum. Please see the edited Fig. 5g.

Minor:

1. Line 50-52, where does the information of sorghum and maize G proteins come from? The listed reference is about the rice G proteins.

> To clearly show the number and classification of crop G-proteins (sorghum, millet, rice and maize), we performed a phylogenetic tree analysis of all searched G-protein homologues (Supplementary Fig. 4a). Thus, we modified this sentence as “Sorghum G protein complex contains one Gα, one Gβ, and two typical and three atypical Gγ subunits, which is similar to rice¹¹ and millet G proteins” (Line 117).

2. Line 64-68, this part should be in introduction rather than in results.

> This part was moved to introduction section (Line 46-50).

3. The figure legend is not clear enough in a number of figures. For example, in Fig. S1a, the abbreviations are not annotated in the legend. In Fig. 4g, the color is not explained in the legend.

> Thanks for your suggestions. We carefully checked all of the figures and figure legends, and added additional details.

4. Line 79-80, How about the effects of other two QTLs?

> Total three loci associated with glume coverage in the natural SAP population were identified. In the diverse sorghum panel of 352 inbred lines, *GCI* explains 9.33% phenotypic variation for glume coverage while the other two loci on chromosome 2 and 3 can explain 10.4% and 7.12% variations respectively. However, *GCI* as a major locus can explain 60.19% glume coverage variation in the F₆ segregated population.

5. There exist different alleles of GC1 with discrepant functions. Therefore, the authors should clearly indicate the exact form of GC1 involved in all experiments of the result part to avoid any confusion.

> We carefully checked the whole text and figures and ensure the correct names of different alleles and constructs used in our manuscript.

Reviewer #3 (Remarks to the Author):

The manuscript by Xie et al. describes the positional cloning and characterization of a major locus regulating reduced glume length present in some domesticated sorghum lines. GWAS identified two major loci controlling glume coverage, and the locus on chromosome 1 (GC1) was fine mapped to a 58 kb region containing 5 genes. A combination of expression analysis and SNP/INDEL association analysis across this region identified a candidate insertion resulting in a frameshift and truncated C-terminal mutation in an atypical G-gamma subunit homologous to a known QTL regulating rice seed size. Multiple independent frame-shift alleles causing a similar deletion in the C-terminal domain all associate with reduced glume length in diverse sorghum accessions. This deletion variant was confirmed by transgenic analysis in both sorghum and millet, which demonstrated that the deletion variant can reduce glume size and is resistant to protein degradation. Interestingly, the C-terminal deletion is

likely a gain of function mutation as knock out CRISPR lines had longer glumes. Gc1 lines with decreased glume size had fewer cells, although the cell size was unaffected suggesting that GC1 affects glume length by inhibiting cell proliferation. Mass-spec analysis identified a patatin-like phospholipase A2 (sbpPLAII-1) protein interactor with GC1. Transient co-expression in tobacco indicate that GC1 accumulation leads to degradation of sbpPLAII-1, and over-expression of sbpPLAII-1 in millet increased glume length, suggesting a possible functional interaction as well. Finally, the authors examined the distribution of GC1 alleles as well as GC1 nucleotide diversity across wild vs. naked grain landraces.

Overall this is an impressive study, presenting a significant amount of work establishing a strong case that C-terminal deletion of GC1 leads to increased protein accumulation and reduced glume length. Although this gene has been identified as a QTL previously in rice, this work shows that an atypical G-gamma influences glume size as well as grain size. In addition, the interaction with a phospholipase provides a novel potential signaling mechanism to follow-up in plant G-protein signaling. While I am enthusiastic about the results, I have a few concerns regarding presentation of the data and its interpretation that should be addressed in a revised manuscript.

> We are very appreciated for the positive evaluation for our manuscript and we modified our manuscript accordingly.

Major Concerns:

1. The expression levels of GC1 vs gc1-a are not consistent from figure 3c to 4a. In 3c it appears that both alleles are expressed at similar levels, while in figure 4a it appears that gc1-a is significantly reduced in expression compared to GC1. Which is correct? The other results are convincing that the C-terminal deletion and it's effects on protein stability are causative. However, if the alleles have different expression levels that could indicate a change in expression pattern related to additional polymorphisms in the regulatory DNA.

> The varied transcriptional levels in the NILs might indicate potential variation in the promoter region of the two natural alleles. We then detected sequence variation in the promoter and found a 341 bp insertion (inse341, see following figure a and b) located at 5'UTR of *gc1-a* allele when compared to *GC1*, despite of other multiple SNPs and small indels. Next, we designed a molecular marker flanking inse341 to verify promoter variation in the selected sorghum accessions used for fig.3c, and found both inse341 variety and gene expression level of *GC1* were not significantly correlated within the five natural *GC1* alleles, as well as glume coverage phenotype (See following figure c).

Second, there is a large range of *GC1* transcript levels (from 0.29 to 5.54 times) among 10 sorghum accessions (used in Fig. 3c) with *GC1* haplotype, see details in Source data Fig. 3c), suggesting that there exists distinct *GC1* expression level in different accessions.

Thus, with these two evidences, we could conclude that both inse341 variety and gene expression level of *GC1* were not related to glume coverage phenotype. To avoid the misunderstanding between Fig. 3c and Fig. 4a, we removed the transcription level of *gc1-a* allele in Fig. 4a. That has no impact for overall results.

Truly as you mentioned that the C-terminal deletion and its effects on protein stability are causative.

2. I was confused by the presentation of the GC1 deletion lines in supplementary figure 10c. Based on these results, it looks to me like the N, G and C domains are required for interaction, since their deletion abolishes BiFC. However, the T-deletion construct maintains a positive BiFC suggesting that it does not contribute to

interaction with PLAII-1. However in the text it says “only the GC1-T domain was sufficient for interaction with SbpPLAII-1” (line 246). I’m not sure what you are trying to express in the text. Are you saying the T domain is sufficient for interaction? If so, that is not consistent with the data. Rather the T-domain was the only domain that did not contribute to interaction.

> Indeed, GC1-T domain was sufficient for interaction with SbpPLAII-1. We are sorry for the confused description in the figure legend of Supplementary Fig. 10b and 10c. The sentence of “...domain-based **deletions** of GC1...” has been changed to “...domain-based **versions** of GC1...” in this new version of manuscript.

4. Figure 5f is critical to the interpretation of the functional interaction of GC1 with sbpPLAII. However, the details of how this experiment was performed were unclear. I assume that these were the results of a co-inoculation of tobacco leaves? If so, how can you reliably modulate GC1 levels while keeping PLAII-1 levels constant? Was this done by altering the relative concentrations of each Agro construct in the inoculation? If so, that seems problematic since expression of each construct is variable across the tobacco cells, and you cannot assess degradation unless PLAII-1 is kept constant.

> It is not a co-inoculation of *Agrobacterium* in tobacco leaves. We add the method in figure legend and so as the detailed description of “Protein degradation assay” in Methods section (Line 525-542). The modified sentences in the legend of Fig. 5e: Total proteins of GFP, GFP-GC1, GFP-gc1-a and SbpPLAII-1 were individually expressed in tobacco leaves. After that, a gradient of concentrations (1/10, 3/10, 5/10 and 10/10 ratio of 100 μ L volume) of GFP-GC1 or GFP-gc1-a protein was incubated with Flag-SbpPLAII-1 protein (10 μ L volume), respectively. Each reaction was topped up by cell lysis from wild type tobacco leaves. GFP protein was used in control reaction.

5. The nucleotide diversity analyses in figures 6b-c show trends consistent with selection. However, there is no statistical analysis of the significance of these findings. Absent this statistical analysis, it is hard to know how to interpret the results.

> Accordingly, we now provide the analysis of nucleotide diversity π (π) value and π ratio with variation sites from genomic segment of *GCI* gene. Also, the statistics of Tajima’ *D* tests, fixation index *Fst* values and HKA tests of *GCI* gene were calculated. A strong positive selection signature with the significantly negative Tajima’s *D*-statistics and a maximum fixation index (*Fst*) value of 0.34 suggests the fixed genetic differentiation between the wild sorghum and naked sorghum cultivars (Line 320-324). Due to the limited picture size of Fig. 6, the selection related analysis has been moved to the additional Supplementary Fig. 13.

Minor concerns

1. Overall the manuscript is clear and easy to read, however some of the sentences are awkward, and the whole manuscript would benefit by careful editing of a native English speaker.

> We carefully checked all of grammar mistakes in the whole text and figure legends.

2. line 85: supplemental figure 2b does not show co-linearity of the M-81E allele with the GWAS results presented in figure 1b. What are the positions of the closest markers in each of these maps?

> The SSR marker *SAM39569* co-separated with *GCI* locus is undoubtedly collinear to the identified locus on Chr.1 from GWAS results. The difference is due to different versions of reference genome. In Supplementary Fig. 2b, left position of the initial map show the physical distance of each marker based on the reference genome of BTx623 (version 3.1). The SSR marker *SAM39569* was located at Chr1: 63,014 Kb (version 3.1) or Chr1: 55,886 Kb (version 1.0). However, the GWAS results were performed based on the genotypes of BTx623 (version 1.0). The leading SNP *SI_55591410* was located at Chr1: 55,591 Kb (version 1.0). Even it is hard to convert position (version 1.0) of SNPs to position (version 3.1) in Fig. 1b and Supplementary Data 2, we preferentially show the latest position (version 3.1) in Supplementary Fig. 2b. We add some description in figure legends.

3. line 103: a C>A SNP creating a stop codon is not a frameshift, please modify the text to clarify the distinct nature of the *gc1-c* allele.

> We corrected this error in the whole text. In line 105, "...four frameshift types..." was replaced by "four truncated types".

4. The *gc1-a* allele appears to be a gain of function. If so, one would predict that it is dominant or semi-dominant over *GC1*. Do you have measurements of a NIL-*gc1-1*/NIL-*GC1* het that would confirm this?

> We have already observed that the *GCI/gc1-a* heterozygous recombinants showed moderate glume coverage between *GCI/GCI* (High glume coverage) and *gc1-a/gc1-a* (Low glume coverage) recombinants in the F₈ generation. That means *gc1-a* is semi-dominant over *GCI*.

5. Lines 216-229: A genome wide transcript profiling experiment was performed, but the results for only a small set of cyclin and a CDK were reported. Is there a good justification for only focusing on these genes, and ignoring all the remaining DEGs? I was surprised that this wasn't directly addressed in the text. Perhaps a supplemental figure containing a broader summary of the RNA-seq would be in order.

> Truly besides those genes in cell cycle and cell proliferation related pathways, we also found numbers of differentially expressed genes (DEGs) enriched, such as transcription factors (TFs), ATP binding, catalytic activity and protein kinase activity pathways. Among them, several *SbMADS* genes were presented but are at different group of rice *OsMADS1*, which acts as a downstream effector of *GS3*. Thus, we did not analyze them in detail but focus on those cell cycle and cell proliferation related pathways due to our physiological data on the cell division. We added a Supplementary Figure 8e and f and a short paragraph in result part (Line 226-233), as well as a discussion about the downstream TFs of *GCI* in the discussion section (Line 348-353).

6. In figure 1c the N values are not given for the SN010 and M-81E lines.

> We added the number (N) of SN010 and M-81E types in Fig. 1c.

7. Figure 2 e, f, and H there are no P values given to assess the significance of the different lines.

> We have already calculated the significant difference by one-way ANOVA with Tukey's multiple comparisons test in Fig. 2e, f and h. The *P*-values are shown in the Source data.

8. Figure 4g the blue vs. orange bars are not labelled.

> We added the labels of the two boxes in Fig. 4g.

9. Figure 6b there is no scale on the X-axis.

> A new picture with X-axis is shown in the Supplementary Fig. 13a.

10. Supp figure 1a: VHGC HGC MGC LGC and VLGC are not defined.

> We added the definition of VHGC HGC MGC LGC and VLGC in the legend of Supplementary Fig. 1.

Reviewers' Comments:

Reviewer #1:

Remarks to the Author:

The revisions are satisfactory! It is acceptable.

Congratulations on the beautiful work!

Reviewer #2:

Remarks to the Author:

The revised manuscript has well addressed the questions and is suitable for acceptance.

A few minor issues might be improved.

Figure 3a, the scale of y axis is not suitable. Figure 3f, left part misses an "(h)".

Figure S7c, there is a mistake in "SIGC1".

Reviewer #3:

Remarks to the Author:

The authors have carefully and thoroughly addressed each of my concerns, including the addition of new data and analyses. I appreciate the thoughtful attention they have given to my suggestions, and the modified manuscript is significantly improved. I look forward to reading it once it is ultimately published. Nice work!

REVIEWERS' COMMENTS

Reviewer #1 (Remarks to the Author):

The revisions are satisfactory! It is acceptable.

Congratulations on the beautiful work!

> We are very appreciated for the positive evaluation for the revised manuscript.

Reviewer #2 (Remarks to the Author):

The revised manuscript has well addressed the questions and is suitable for acceptance.

A few minor issues might be improved.

Figure 3a, the scale of y axis is not suitable. Figure 3f, left part misses an “(h)”.

Figure S7c, there is a mistake in “SIGC1”.

> We are very appreciated for the positive evaluation for the revised manuscript. We modified these minor issues in the figures accordingly.

Reviewer #3 (Remarks to the Author):

The authors have carefully and thoroughly addressed each of my concerns, including the addition of new data and analyses. I appreciate the thoughtful attention they have given to my suggestions, and the modified manuscript is significantly improved. I look forward to reading it once it is ultimately published. Nice work!

> We are very appreciated for the positive evaluation for the revised manuscript.